# Technical Note: Comparison of radiometric techniques for estimating recent organic carbon sequestration rates in inland wetland soils

Purbasha Mistry[1], Irena F. Creed[1,2], Charles G. Trick[3], Eric Enanga[2], David A. Lobb[4]

[1] School of Environment and Sustainability, University of Saskatchewan, 117 Science Place, Saskatoon, SK, S7N 5C8, Canada
[2] Department of Physical and Environmental Sciences, University of Toronto, 1265 Military Trail, Toronto, ON, M1C 1A4, Canada
[3] Department of Health and Society, University of Toronto, 1265 Military Trail, Toronto, ON, M1C 1A4, Canada
[4] Department of Soil Science, University of Manitoba, 13 Freedman Crescent, Winnipeg, MB, R3T 2N2, Canada

*Correspondence to:* Irena F. Creed (irena.creed@utoronto.ca)

**Abstract.** For wetlands to serve as natural climate solutions, accurate estimates of organic carbon (OC) sequestration rates in wetland sediments are needed. Dating using cesium-137 ($^{137}$Cs) and lead-210 ($^{210}$Pb) radioisotopes is commonly used for measuring OC sequestration rates in wetland sediments. $^{137}$Cs radioisotope dating is relatively simple, with calculations based on a single point representing the onset (1954) or peak (1963) of the $^{137}$Cs fallout. $^{210}$Pb radioisotope dating is more complex as the calculations are based on multiple points. Here, we show that reliable dating of sediment cores collected from wetlands can be achieved using either $^{137}$Cs or $^{210}$Pb dating or their combination. However, $^{137}$Cs and $^{210}$Pb profiles along the depth of sediment cores need to be screened, analyzed, and interpreted carefully to estimate OC sequestration rates with high precision. To this end, we propose a decision framework for screening $^{137}$Cs and $^{210}$Pb profiles into high- and low-quality sediment profiles, and we compare dating using the 1954 and 1963 time-markers, i.e., the rates of sedimentation and, consequently, OC sequestration over the past ~60 years. Our findings suggest that $^{137}$Cs- and $^{210}$Pb-based OC sequestration rates are comparable, especially when using the 1963 (vs. 1954) time-marker.

## 1 Introduction

Wetlands in agricultural landscapes serve a crucial role in providing habitat for wildlife, regulating climate, improving water quality, and reducing floods. Moreover, these wetlands have the potential to sequester organic carbon (OC) (Bridgham et al., 2006; Nahlik and Fennessey, 2016; Bansal et al., 2023). Accounting for the balance between the sequestration and emission of carbon can help establish wetlands as essential candidates for natural climate solutions by offsetting carbon emissions (Hambäck et al., 2023). These wetlands embedded in agricultural landscapes are recognized as temperate inland wetland soils. The global carbon stock of temperate inland wetland soils is estimated to be 46 Pg C to 2 m depth, and Canada's temperate inland wetland soils are estimated to contain 4.6 Pg C (Bridgham et al., 2006). Compared to peatlands, the rapid

rate of OC sequestration and the more considerable spatial extent of temperate inland wetland soils can help contribute
significantly to regional or national carbon sequestration (Bridgham et al., 2006; Nahlik and Fennessey, 2016).
Canada encompasses around 25% of the world's wetlands, with an area of approximately 1.29 million square kilometers,
which accounts for 13% of the country's terrestrial area (Environment and Climate Change Canada, 2016), highlighting the
global importance of these wetlands. Unfortunately, there is minimal data on the OC sequestration rates in these wetlands.
To estimate the OC sequestration potential of these wetlands, it is essential to establish precise measurements to quantify
wetland OC sequestration, develop strategies to promote conservation and restoration efforts, incorporate carbon credits in
the carbon markets, and validate the wetland-based ecosystem services.

There are several ways to estimate the potential of wetlands to store OC (Bansal et al., 2023). One of these methods is
radiometric dating, which can calculate the OC storage rates of wetlands over periods of 10 to $\geq$ 1,000 years. Frequently used
radioisotopes for radiometric dating are cesium-137 ($^{137}$Cs) and lead-210 ($^{210}$Pb), which can be used to estimate relatively
recent (up to the last 100 years) OC sequestration rates (Villa and Bernal, 2018). Estimating OC sequestration rates involves
building an age-depth profile or model of $^{137}$Cs and $^{210}$Pb from sediment cores that demonstrate the relationship between the
depth of sediment layers and their corresponding age. Since the inorganic radioisotopes ($^{137}$Cs and $^{210}$Pb) strongly bind with
the soil particles once in contact, the radioisotopes can act as an efficient tracer for investigating OC sequestration rates
(Ritchie and McHenry, 1990; Craft and Casey, 2000). These characteristics allow for accurate tracking of carbon movement
within ecosystems, thereby enabling the extraction of detailed information about carbon sequestration dynamics in wetlands.

The characteristics of $^{137}$Cs and $^{210}$Pb to estimate wetland OC sequestration rates are presented in Table 1. $^{137}$Cs is an
artificial radioisotope that was produced during thermonuclear bomb testing in the 1950s and 1960s, with the onset of
atmospheric deposition in 1954 and a peak in 1963 (Ritchie and McHenry, 1990). The testing caused radioactive uranium to
decay, and, as a result, $^{137}$Cs isotope was released into the atmosphere, which was then deposited around the globe. Although
there may be challenges in applying our study to some parts of the world, the information is generally applicable and
valuable for consideration in all regions. We encourage others to customize this approach further for use in other regions
where Cs deposition histories vary.

$^{137}$Cs has a half-life of 30.17 years, which can be used to estimate the last ~50-70 years of OC sequestration rates in wetlands
(e.g., Bernal and Mitsch, 2012). $^{137}$Cs dating assumes constant sedimentation rates measured since 1954 or 1963. In using the
two time-markers for $^{137}$Cs, we do not expect the sedimentation rates to be equal, but we do expect them to be similar. The
onset and the peak of $^{137}$Cs activity at 661.6 keV can be used to mark 1954 and 1963, respectively. These time-markers
(1954 and 1963) can date sediment layers (Pennington et al., 1973; Ritchie and McHenry, 1990; DeLaune et al., 2003) and
consequently the OC sequestration rates. $^{137}$Cs has an additional time-marker for Europe in 1986 due to the Chernobyl
nuclear accident and for Japan in 2011 due to the Fukushima Daiichi nuclear accident (Foucher et al., 2021), indicating that
OC sequestration estimates can be derived for different timescales. In the Americas, we do not see evidence of the 1986 or
2011 $^{137}$Cs peak, which is observed in Europe and Japan, respectively, so we did not need to use other radioisotope
techniques (e.g., $^{239+240}$Pu) to distinguish the 1986 or 2011 $^{137}$Cs peak from the 1963 $^{137}$Cs peak. $^{137}$Cs dating requires a
gamma spectrometer to estimate OC sequestration rates. Sample preparation for gamma analysis involves drying, weighing,
disaggregating, homogenizing, and sieving (Bansal et al., 2023). Samples vary from 1 to 1,500 g, with smaller samples
associated with higher uncertainties and, therefore, requiring longer times to analyze. Gamma analysis counting times range
from 4 to 48 h for each sample (e.g., 4 to 12 h in Li et al., 2007; 12 to 24 h in Zarrinabadi et al., 2023; and 24 to 48 h in
Kamula et al., 2017). $^{137}$Cs dating provides a simple result (an average sedimentation rate), while $^{210}$Pb dating provides a
more complex result (using a supply rate model to reveal trends in sedimentation rates). Plutonium (Pu) may replace $^{137}$Cs in
the future due to concerns of half-life and persistence as a dating tool. In essence, $^{239+240}$Pu has the same source and
deposition mechanism as $^{137}$Cs. Its longer half-life will make its peak measurable when $^{137}$Cs is no longer measurable.

Unlike $^{137}$Cs, $^{210}$Pb is a naturally occurring radionuclide derived from $^{238}$U and deposits atmospherically from the decay of
radium-226 ($^{226}$Ra) (Walling and He, 1999). $^{210}$Pb has a half-life of 22.3 years and is used to estimate the last 10-150 years of
OC sequestration rates in wetlands (Craft and Richardson, 1998; Craft and Casey, 2000; Craft et al., 2018; Creed et al.,
2022). $^{210}$Pb activity can be measured using gamma (observed at 46.5 keV) and alpha spectrometry (destructive) (Walling
and He, 1999; Bellucci et al., 2007). Traditional alpha analysis requires 0.2-0.5 g of sample and additional sample
preparation involving leaching with hydrochloric and nitric acid and electroplating (up to 24 h for sample preparation)
(Bansal et al., 2023). Alpha analysis can be considered an indirect method for $^{210}$Pb dating where polonium-210 ($^{210}$Po)
activity is measured, assuming both $^{210}$Pb and $^{210}$Po are in a secular equilibrium. $^{210}$Pb activity is calculated by comparing
$^{210}$Po activity against the known activity of $^{209}$Po (isotope tracer). In alpha analysis, the additional time required for sample
preparation is compensated by running multiple samples simultaneously (Bansal et al., 2023). Gamma and alpha
spectrometry of $^{210}$Pb provides the total $^{210}$Pb activity, which incorporates unsupported (or excess) $^{210}$Pb ($^{210}$Pb$_{ex}$) and
supported $^{210}$Pb. $^{210}$Pb$_{ex}$ is used to determine the mass or sediment accumulation rate. Supported $^{210}$Pb is derived from the
natural decay of $^{226}$Ra in the sediment, while unsupported $^{210}$Pb comes from the decay of atmospheric radon-222 ($^{222}$Rn),
which deposits $^{210}$Pb onto the sediment surface from the air. Unsupported $^{210}$Pb activity decreases over time due to
radioactive decay, unlike supported $^{210}$Pb (Appleby and Oldfieldz, 1983). The choice of model used in $^{210}$Pb dating can
reflect constant and variable sedimentation rates (Sanchez-Cabeza and Ruiz-Fernandez, 2012) and, consequently, OC
sequestration rates in wetlands. Some models used for $^{210}$Pb dating are (1) constant flux-constant sedimentation (CFCS)
model, (2) constant rate of supply (CRS) model, and (3) constant initial concentration (CIC) model (Appleby and Oldfield,
1978). Both [137]Cs and [210]Pb provide suitable time-markers and a longer time horizon compared to direct measurements using
the time-marker of horizons (2-10 years) to study sediment accretion and, subsequently, OC sequestration rates in wetlands
(Bernal and Mitsch, 2013; Villa and Bernal, 2018). In this study, we compared the average OC sequestration rate derived
from [137]Cs temporal markers with the progressive OC sequestration rates derived using a constant rate of supply model
applied to [210]Pb.
**Table 1: Characteristics of $^{137}$Cs and unsupported $^{210}$Pb ($^{210}$Pb$_{ex}$) dating to estimate sedimentation rates in wetlands.**

| Method of radiometric dating | $^{137}$Cs | $^{210}$Pb$_{ex}$ |
|---|---|---|
| Type of radioisotope | Artificial (atmospheric deposition 1954 – 1963). | Natural. |
| Half life | 30.17 years. | 22.3 years. |
| Time-marker | 1954 (onset) and 1963 (peak). | Recent (10-20 years) to a maximum of 50-150 years. |
| Radiometric Technique | Gamma spectrometry (nondestructive). | Gamma (nondestructive) and alpha spectrometry (destructive). |
| Pre-processing | Drying, weighing, disaggregating, homogenizing, and sieving. | For gamma analysis, drying, weighing, disaggregating, homogenizing, and sieving prior to analysis on a gamma counter. |
| | | For alpha analysis, leaching with hydrochloric and nitric acid and electroplating of $^{210}$Po which constitutes allowing the digested and therefore extracted $^{210}$Po isotope solution to settle on silver coins overnight before measuring the $^{210}$Po (known tracer) and $^{210}$Po activity (sample) next morning through the alpha counter/ensemble. |
| Sample size | Minimum 1 g (larger sample size has higher certainty). | 1 to 5 g for gamma spectrometry, 0.2 to 0.5 g for alpha spectrometry. |
| Time requirement for radiometric dating | 48 h for each sample for gamma spectrometry. | 48 h for each sample for gamma spectrometry. |
| | | 48 to 72 h for multiple samples plus sample preparation time per multiple samples. |
| Output | A single average sedimentation rate. | Variable sedimentation rate. |
| Estimation approach | Onset of $^{137}$Cs activity represents 1954 and highest peak of $^{137}$Cs activity represents 1963, observed at 661.6 keV. | Activity of $^{210}$Pb is observed at 46.5 keV. Excess $^{210}$Pb is used to determine the vertical accretion. |
| Complexity in estimation | Simple; estimated by using time-marker of onset or peak $^{137}$Cs activity and associated sediment accumulation. | More complex; estimated by one of several models to estimate sedimentation rate. Most common models are (1) constant flux–constant sedimentation model, (2) constant rate of supply model, and (3) constant initial concentration model (Appleby and Oldfield, 1978) |


The combined use of $^{137}$Cs and $^{210}$Pb may improve the accuracy of the dating estimation (Drexler et al., 2018; Creed et al.,
2022). The more detailed assessment accrues a higher cost and time requirement, and the need for specialized equipment and
technical expertise to conduct laboratory and data analyses may constrain the research efforts (Bansal et al., 2023).
Furthermore, factors such as timescales, analytical complexity in interpreting radioisotope profiles (e.g., $^{137}$Cs peak clarity),
variability in atmospheric deposition, and mobilization of radioisotopes can contribute to uncertainty (Drexler et al., 2018;
Loder and Finkelstein, 2020; Zhang et al., 2021; Bansal et al., 2023) and limit the applicability of one radioisotope over the

other. Therefore, it is essential to consider the advantages and potential challenges of using radioisotopes before designing research studies.

The main objective of this research paper is to explore the use of $^{137}$Cs and $^{210}$Pb to estimate recent OC sequestration rates in undisturbed (i.e., not directly impaired by human activities) temperate inland wetland soils located on agricultural landscapes. Here, we aim to (1) categorize $^{137}$Cs or $^{210}$Pb profiles into high- and low-quality via a decision framework, (2) apply the decision framework to estimate OC sequestration rates, (3) use 1963 and 1954 time-markers to compare the $^{137}$Cs- and $^{210}$Pb-based OC sequestration rates to get a better understanding of the sediment history, and (4) select the best approach for $^{137}$Cs and $^{210}$Pb to estimate the OC sequestration rates with highest precision. This study helps reduce uncertainty in studies that rely on $^{137}$Cs or $^{210}$Pb radioisotope dating.

## 2      Methods

### 2.1      Sediment core collection

Triplicate sediment cores were collected from 30 undisturbed temperate inland wetland soils in agricultural landscapes across southern Canada (Fig. 1). A summary of the physical characteristics of these wetlands can be found in Supplementary Table 1. These wetlands were undisturbed, with no known history of cultivation. The sediment cores were extracted from the center of the wetland, constituting the open-water area. A Watermark Universal Corer (inner diameter of 6.8 cm) or VibeCore Mini with poly core tubes (inner diameter of 7.6 cm) were used to collect most of the sediment cores. A JMP BackSaver Soil sampler (inner diameter of 3.8 cm) was used for compacted sediment cores. Shallow (15 to 90 cm) sediment cores were sectioned into 1- or 2-cm increments. Deeper (> 90 cm) sediment cores were sectioned into 5-cm increments. The sediment cores were stored at -5 °C for further processing at the laboratory.

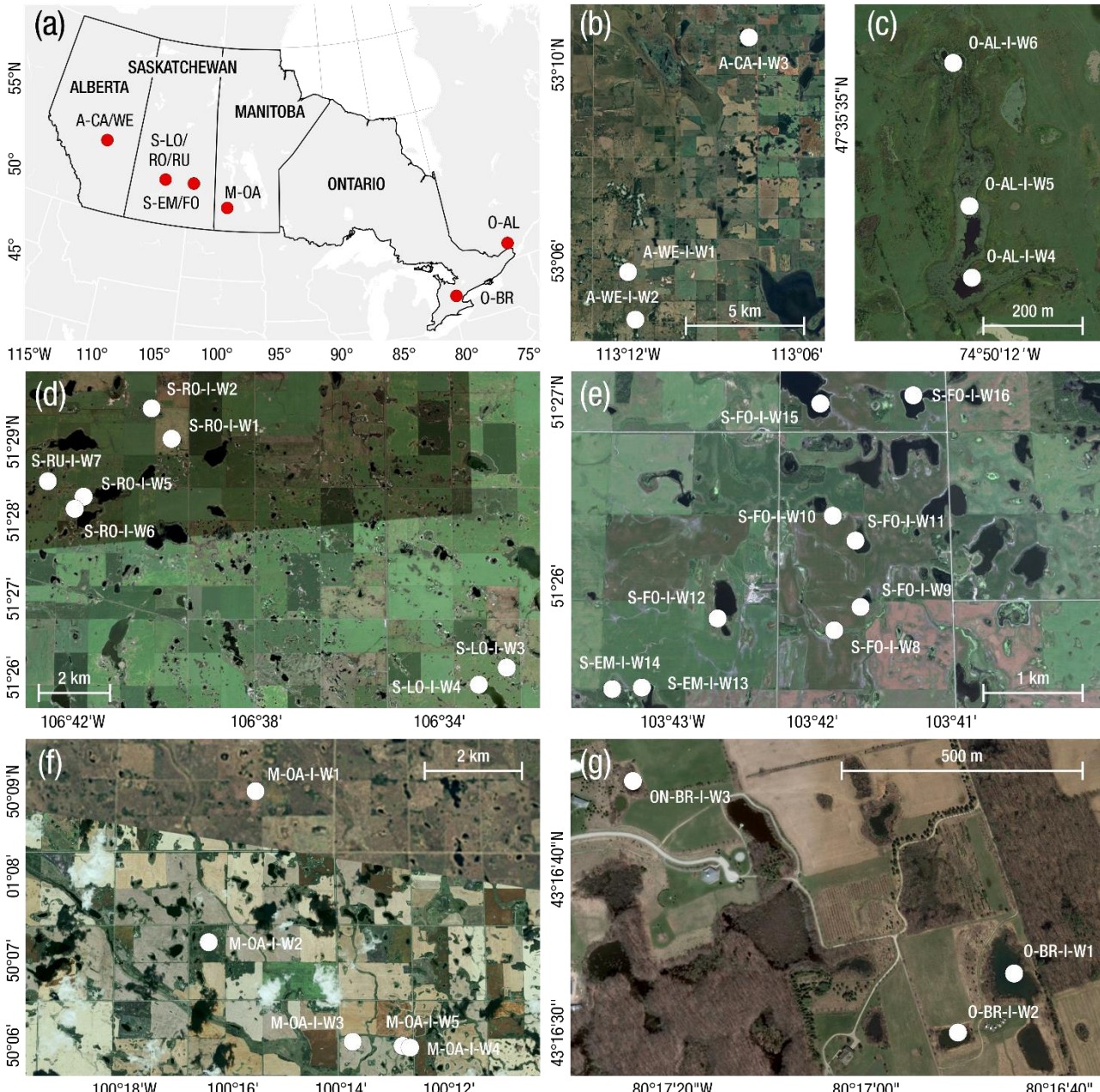

Figure 1: (a) Study area situated in four provinces of Canada; (b) three wetland sites in Alberta (AB); (c) three wetland sites in Ontario (ON); (d) seven wetland sites in Saskatchewan (SK); (e) nine wetland sites in Saskatchewan (SK); (f) five wetland sites in Manitoba (MB); and (g) three wetland sites in Ontario (ON). Figures (b)-(g) are based on the sampling locations of wetlands used in this study reproduced using Google Earth Images [(b) and (c) ©2024 Airbus; (d), (e), and (g) ©2024 Maxar Technologies; (f) ©2024 Airbus and Maxar Technologies].

## 2.2    Generation of $^{137}$Cs and $^{210}$Pb profiles

Sediment core increments were weighed (wet mass), dried, weighed again (dry mass), disaggregated, homogenized, and sieved. The increments were sieved to remove gravel (> 2 mm); radioisotopes do not bind on the gravel, and gravel does not contain OC; therefore, eliminating gravel improves the estimate of radioisotopes and OC. The increments were counted at 661.6 keV for $^{137}$Cs activity and 46.5 keV for $^{210}$Pb activity. $^{137}$Cs analysis was performed using a gamma spectrometer, and $^{210}$Pb analysis was performed using both gamma and alpha spectrometers to increase throughput rates. The gamma analysis was conducted using the high-purity germanium detectors; e.g., Broad Energy Germanium detectors (BE6530) and Small Anode Germanium well detectors (GSW275L) (Mirion Technologies, Inc., Atlanta, GA, USA). The alpha analysis was conducted using ORTEC® alpha spectrometer (AMETEK® Advanced Measurement Technology, TN, USA). Both radioisotope analyses were performed at the Landscape Dynamics Laboratory, University of Manitoba, Winnipeg, Canada. Although the underlying principles of gamma and alpha analysis differ, each focuses on quantifying the decay of $^{210}$Pb, generating comparable results (Zaborska et al., 2007). Measurement accuracy of gamma detectors is ensured by assessing the counting errors with reference materials within the same geometry as the sample (e.g., petri dish). Detection error was < 10% with a counting time of up to 24 h. Furthermore, Landscape Dynamics Laboratory undergoes regular Proficiency Testing through the International Atomic Reference Material Agency (IARMA) and previously through the International Atomic Energy Agency (IAEA) to ensure acceptable accuracy and precision of analytical results using gamma spectroscopy.

## 2.3    Screening of $^{137}$Cs and $^{210}$Pb profiles

Sediment cores were screened to remove profiles with evidence of vertical mixing, and then the remaining profiles were used to estimate OC sequestration rates using $^{137}$Cs or $^{210}$Pb radioisotope dating. The actual $^{137}$Cs peak can vary from the expected peak, increasing uncertainty in $^{137}$Cs dating (Drexler et al., 2018). $^{137}$Cs peaks can be "noisy" or "disturbed"; i.e., flattened, broadened, truncated, mixed, fluctuating (Drexler et al., 2018), or one-sided where the $^{137}$Cs peaks appear at the surface of the sediment core (indicating no or little sedimentation since 1963). The magnitude and shape of the $^{137}$Cs peaks observed in the sediments can be affected by the atmospheric deposition rate of $^{137}$Cs, which is obviously affected by the number and magnitude of emission events and the weather conditions following these events (UNSCEAR, 2000). The magnitude and shape of these peaks are also impacted by the movement of water and sediment within each wetland's catchment during the peaks' development (Milan et al., 1995; Zarrinabadi et al., 2023). Here, changes in the shape of the peaks are caused by the upward and downward movement of the sediment within the sediment profile (the movement of $^{137}$Cs through diffusion (Klaminder et al., 2012) is presumed negligible). Bioturbation can cause an upward and downward mixing of the $^{137}$Cs in the profile, resulting in peak attenuation (Robbins et al., 1977). Even wave action during the period of atmospheric deposition will have a similar attenuation effect (Andersen et al., 2000; Zarrinabadi et al., 2023). Following peak atmospheric deposition, soil erosion and the accumulation of sediment will deliver sediments to the top of the profile, and those sediments

may be higher or lower in concentration depending on the degree of preferential sediment transport and the associated enrichment or depletion of $^{137}$Cs in the added sediment (Zarrinabadi et al., 2023). Such noise in $^{137}$Cs peaks needs careful interpretation to avoid over- or under-estimating the OC sequestration rates.

*Selecting suitable cores:* Of the 90 sediment cores (30 wetlands x 3 replicates = 90), 79 were suitable (complete and datable) for $^{137}$Cs dating and 47 for $^{210}$Pb dating. Only some replicates from the same wetland were ideal for interpretation or further screening. The suitability of $^{137}$Cs profiles for dating was assessed by zero activity before the onset and peak of $^{137}$Cs activity. The suitability of $^{210}$Pb profiles for dating was evaluated by determining the exponential decline in $^{210}$Pb activity with depth until background levels are reached.

*Classification of the selected $^{137}$Cs profiles:* The 79 suitable $^{137}$Cs profiles were then classified into high- and low-quality using the following steps (Fig. 2a):

1. The $^{137}$C depth profile and the shape of the peak were assessed. A clear and distinct peak associated with several points on both sides of the peak verified the $^{137}$Cs depth profile as high-quality (e.g., Fig. 3a).

2. When analyzing sediment samples, a clear peak in $^{137}$Cs activity didn't always exist (e.g. Fig 4a). If a peak was absent, which could have resulted from sediment influxes with very high or very low $^{137}$Cs activity levels, the total $^{137}$Cs activity of the entire profile was examined. If the cumulative $^{137}$Cs inventory value for the entire profile was $\geq$ 500 Bq m$^{-2}$, then the $^{137}$Cs profile was considered high-quality. Conversely, if the cumulative $^{137}$Cs inventory value for the entire profile was < 500 Bq m$^{-2}$, the $^{137}$Cs profile was considered low-quality. The cutoff cumulative $^{137}$Cs inventory value of 500 Bq m$^{-2}$ was established by assessing the $^{137}$Cs reference inventory value, the value of $^{137}$Cs present in a non-eroded system with an undisturbed profile. The $^{137}$Cs reference inventory value differs from region to region (Owens and Walling, 1996), and the most proximal regional value was used to select the cutoff $^{137}$Cs inventory value (Sutherland, 1991; Kachanoski and Von Bertoldi, 1996; Zarrinabadi et al., 2023). The $^{137}$Cs reference inventory is a catchment-wide reference value and not specific to the wetland center; thus, the cumulative $^{137}$Cs inventory value of 500 Bq m$^{-2}$ was viewed as a conservative indicator of the suitability of the $^{137}$Cs profiles. Ideally, reference sites are large, open, level, non-eroded areas, usually in forage or grassland since the 1950s, and within 10 km of the site of interest. In this study, it was impossible to identify a suitable reference site near every wetland; it is usually difficult to find reference sites in agricultural landscapes. However, we could locate reference sites used in other studies within 50 km except from nine wetlands in SK (51° N and 104° W), which were ~150 km from the reference site. Although this was not considered ideal, it was considered acceptable.

*Classification of the selected* [210]Pb *profiles:* The 47 suitable [210]Pb profiles were classified into high- and low-quality profiles
based on the following steps (Fig. 2b):

1.   [210]Pb activity were plotted with a log-transformed [210]Pb$_{ex}$ against mass depth (g cm$^{-2}$).
2.   A linear regression analysis was performed (where the slope is used to derive the mass or sediment accumulation

rate in g cm$^{-2}$ yr$^{-1}$).

3.   If the linear regression passed both normality and constant variance tests and had an $R^2 > 0.5$ and a p-value $< 0.05$,

then the [210]Pb profile was classified as high-quality (e.g., Fig. 3a).

4.   If either normality and constant variance tests were not passed, with an $R^2 \leq 0.5$ or a p-value $\geq 0.05$, then the [210]Pb

profile was considered low-quality (e.g., Fig. 3b).


The [210]Pb profiles were also classified using a two-step piecewise linear regression model to capture recent shifts in OC
sequestration rates. However, no significant improvement was observed. Consequently, [210]Pb-based OC sequestration rates
were derived from the linear regression line. An $R^2 > 0.5$ was selected as the cut-off for selecting high-quality over low-
quality profiles. Increasing the cut-off $R^2$ value may produce better profiles to be chosen for the study. Still, it can reduce the
number of available sediment cores and potentially ignore the natural variability and significant events occurring in the real
environment.

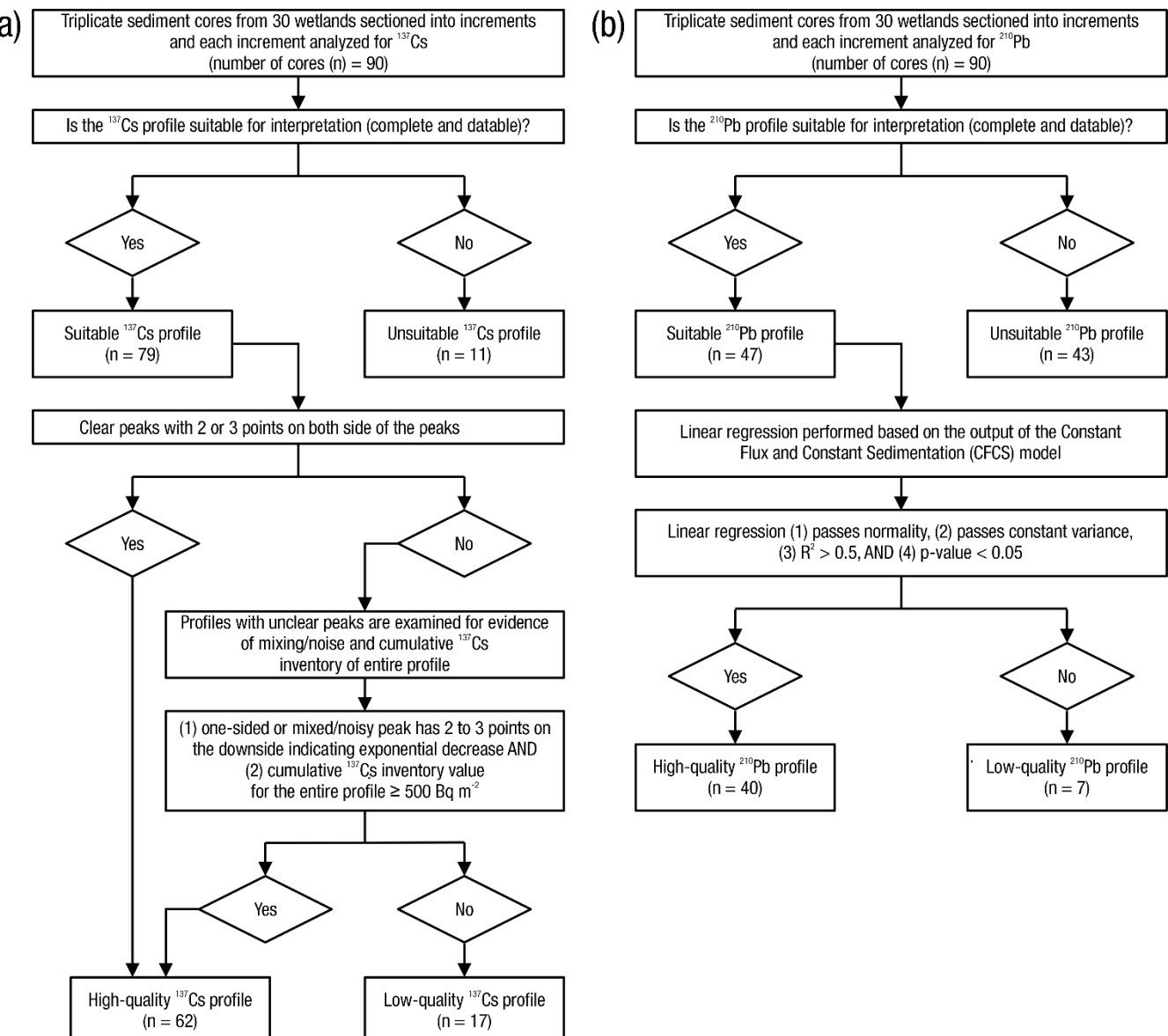


Figure 2: Classification of high- and low-quality [137]Cs and [210]Pb profiles outlining the decision frameworks for screening (a) [137]Cs and (b) [210]Pb profiles.

**2.4     Organic carbon stocks and sequestration rates**
Radioisotope activity measurements were utilized to assign two time-markers, one for 1954 and the other for 1963, in the
sediment cores. Sediment radioisotope dating was used to calculate the rates of sediment or mass accumulation and OC
sequestration.

For $^{137}$Cs dating, sediment accumulation and OC sequestration rates (Mg ha$^{-1}$ yr$^{-1}$) were estimated using the cumulative sum
of sediment or OC (Mg ha$^{-1}$) from the surface to the depth corresponding to the time-markers of $^{137}$Cs of each core and
dividing by the number of years from the time-marker to the years the samples were collected. Unit conversion is applied to
report the OC sequestration rate estimates in Mg ha$^{-1}$ yr$^{-1}$ from g cm$^{-2}$ yr$^{-1}$ for easy standardization and comparability with
other studies. For $^{137}$Cs profiles with noisy peaks and comparatively larger cumulative $^{137}$Cs inventory values, the first
elongated peak with a sharp rise after the onset of the $^{137}$Cs peak was considered the 1963 peak instead of the peak with the
highest activity in the profile.

For $^{210}$Pb dating, mass or sediment accumulation and OC sequestration rates were estimated using the Constant Flux and
Constant Sedimentation (CFCS) model (Sanchez-Cabeza and Ruiz-Fernandez, 2012; Kamula et al., 2017). Here, $^{210}$Pb$_{ex}$ was
estimated by subtracting $^{226}$Ra activity (186 keV) from the total $^{210}$Pb activity. The CFCS model uses the log-linear
relationship of $^{210}$Pb$_{ex}$ with mass depth and converts $^{210}$Pb$_{ex}$ to the mass or sediment accumulation rate and, consequently, the
OC sequestration rate. The OC stock was estimated by taking the cumulative sum of OC (Mg ha$^{-1}$) from the surface of each
sediment core to the depth increments represented by the time-marker (e.g., 1963).

OC stocks for the 1954 and 1963 time-markers were calculated by multiplying the OC content per unit mass of soil (g).
Here, OC content was calculated from OC concentration (%) measured by loss-on-ignition (LOI) method (Kolthoff and
Sandell, 1952; Dean, 1974) by the mass of sediment for each section interval and specific depth interval per unit area (g cm$^{-2}$
$^{2}$) down the profile to the respective time-marker. OC (%) was calculated by multiplying organic matter (%) by LOI with
0.58, assuming 58% of the organic matter is carbon. Despite the broad applicability, simplicity in measurement techniques,
and cost-effectiveness, the LOI approach is associated with some limitations, such as the ignition of non-organic particles at
high temperatures or the use of a conventional conversion factor (Pribyl, 2010; Hoogsteen et al., 2015), which can result in
over-estimation of OC content.
**2.5     Statistical analysis**
Statistical analyses used sediment cores with $^{137}$Cs- and $^{210}$Pb-based OC sequestration rates available (number of sediment
cores (n) = 44). The $^{137}$Cs—and $^{210}$Pb-based estimates of OC sequestration rates were compared using a quantile-quantile (Q-

Q) plot. First, the comparison was done via assessment of the Q–Q plots. Four sample datasets were used to construct Q-Q plots to compare the distribution of $^{137}$Cs- and $^{210}$Pb-based OC sequestration against the 1:1 line.

The sample datasets included:

- D1, all suitable $^{137}$Cs and $^{210}$Pb profiles with OC sequestration rates estimated since 1954 (n = 44).

- D2, all suitable $^{137}$Cs and $^{210}$Pb profiles with OC sequestration rates estimated since 1963 (n = 44).

- D3, high-quality $^{137}$Cs and $^{210}$Pb profiles with OC sequestration rates estimated since 1954 (n = 30).

- D4, high-quality $^{137}$Cs and $^{210}$Pb profiles with OC sequestration rates estimated since 1963 (n = 30).

A Q-Q plot was calculated for each dataset. The x- and y- coordinates of a point in a Q-Q plot corresponded to the $p^{th}$ percentiles of the two OC sequestration rate estimates being compared in the plot. Here, $p = (k – 0.5) / n$, where n is the sample size and k = 1, …, n (Jain et al., 2007). The distribution of the points on the Q-Q plot was compared against the $y = x$ (1:1) line to assess whether the two OC sequestration rate estimates are similar. If the points were distributed in a straight line and close to a 1:1 line, then it suggested that the two estimates came from the same distribution. In contrast, if the points were not distributed in a straight line or deviated from the 1:1 line, then it suggested that the two estimates did not come from the same distribution. The Q-Q plots were generated in Microsoft Excel (Microsoft 365, Version 2402, Redmond, WA).

Since interpreting the Q-Q plot through a visual inspection can be subjective to human perception, we compared the $^{137}$Cs- and $^{210}$Pb-based OC sequestration rate estimates using a distance sampling model. A distance sampling model captures how the detectability of objects from the observer (walking along a straight line) decreases with the increase in the object-to-observer distance. If the objects are closely distributed along the observer's path (i.e., if points of the Q-Q plot were closely distributed along the 1:1 line), then the distribution of the distances is expected to be a half normal distribution. The Cramer-von Mises test was used to estimate whether the distances (q1, q2, …, qn) from the points to the 1:1 line were from a half-normal distribution. Given a set of distance samples (q1, q2, …, qn) and a detection function, the Cramer-von Mises test builds a model that fits the distance sampling data to the detection function (for details on modelling, see Miller et al., 2019). A half-normal key is commonly used as a detection function, corresponding to a half-normal distribution's shape.

The Cramer-von Mises test produced a p-value and Akaike's Information Criterion (AIC) as its test statistic. A p-value larger than the significant level (p = 0.05) indicated that the likelihood of points being observed closer to the 1:1 line is high and

that the probability decreases as the distances increase. This provided evidence of the points being closely distributed along
the 1:1 line. The AIC was used to rank the distance sampling models, which are built by the Cramer-von Mises test, from
best to worst (e.g., Burnham and Anderson, 2003); a small AIC value indicates a good fit to the half-normal key and thus
provides evidence that the points are close to the 1:1 line (Miller et al., 2019). The distance sampling Cramer-von Mises test
was computed using the "distance" package in R version 4.0.3 (Miller and Clark-Wolfe, 2023; R Core Team, 2023).

## 284 3 Results

### 285 3.1 High- and low-quality $^{137}$Cs and $^{210}$Pb profiles

Of the 79 suitable $^{137}$Cs profiles, 62 (78%) were classified as high-quality. Of the 62 high-quality $^{137}$Cs profiles, 61% had
clear and distinct peaks, with a smooth rise and decline. In contrast, the remaining 39% had noise—either one-sided peaks or
disturbed peaks (e.g., Fig. 4). Of the 62 high-quality $^{137}$Cs profiles, 4 (6.5%) were repositioned to capture the $^{137}$Cs enriched
sediments post 1963 (e.g., $^{137}$Cs profile of S-LO-I-W4-T2-CW-R2 in Supplementary Fig. 2a). In these profiles, which had a
cumulative $^{137}$Cs inventory value > 1,200 Bq m$^{-2}$, the depth that corresponded to $^{137}$Cs cumulative inventory value of ~500
Bq m$^{-2}$ was considered as the 1963 time-marker. The high total quantities of $^{137}$Cs profile inventories can be attributed to
receiving $^{137}$Cs enriched sediments from the surrounding landscape. Sediments that have undergone substantial preferential
detachment and entrainment on their pathway into a wetland can have very high concentrations of $^{137}$Cs and, when
interlayered with sediments that are not so enriched, can generate multiple $^{137}$Cs peaks in the sediment profile peaks after
1963. These observed multiple peaks are local and not regional, ruling out the association with Chernobyl and Fukushima
events. Two $^{137}$Cs profiles were considered high-quality despite a cumulative $^{137}$Cs inventory value < 500 Bq m$^{-2}$ because the
1963 peak was clear, distinct, and elongated with two-to-three points on both sides of the peak (e.g., $^{137}$Cs profile of M-OA-
I-W4-T2-CW-R2 in Supplementary Fig. 7b). One $^{137}$Cs profile was considered high-quality despite showing marginal
quality to the set criteria in the decision framework, where the peak profile had good shape with several points on the
downside of the peak and one point on the other side and had a cumulative $^{137}$Cs inventory value of 499 Bq m$^{-2}$. One $^{137}$Cs
profile was classified as low-quality despite a cumulative $^{137}$Cs inventory value > 500 Bq m$^{-2}$ because the peak was highly
fluctuating and not discernible (e.g., $^{137}$Cs profile of O-AL-I-W6-T1-CW-R1 in Supplementary Fig. 12b).

Of the 47 $^{210}$Pb profiles, 40 (85%) were classified as high-quality.

There were 44 sediment cores with both $^{137}$Cs and $^{210}$Pb suitable profiles available. Of these, 30 were categorized as high-
quality $^{137}$Cs and high-quality $^{210}$Pb (Fig. 3a), six were categorized as high-quality $^{137}$Cs and low-quality $^{210}$Pb (Fig. 3b),
seven were classified as low-quality $^{137}$Cs and high-quality $^{210}$Pb (Fig. 4a), and one was categorized as low-quality $^{137}$Cs and
low-quality $^{210}$Pb (Fig. 4b). (See Supplementary Figs. 1 to 12 for $^{137}$Cs and $^{210}$Pb profiles in all study wetlands.)

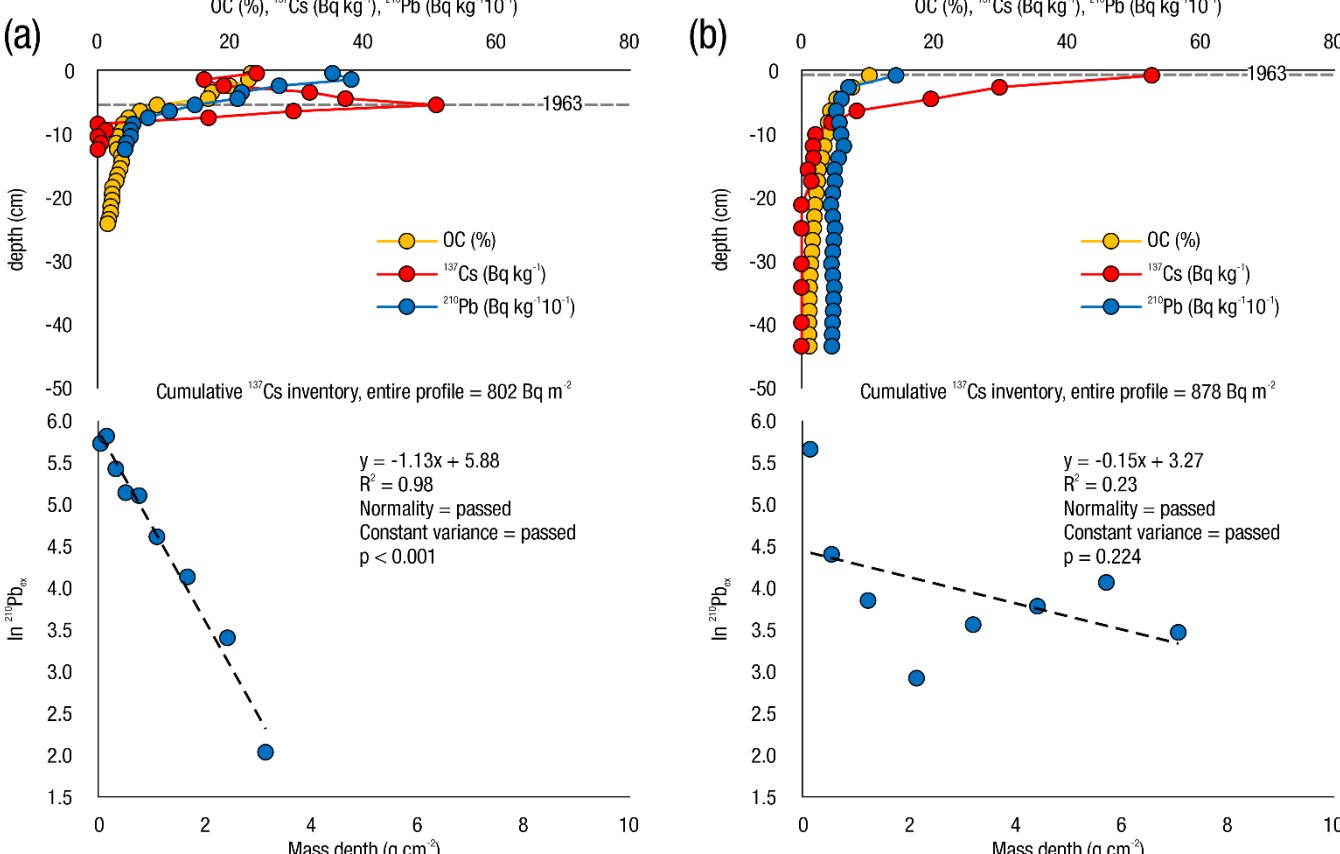

**Figure 3: Examples of $^{137}$Cs and $^{210}$Pb classifications showing OC (%), $^{137}$Cs (Bq kg$^{-1}$), and $^{210}$Pb (Bq kg$^{-1}$) depth profiles and plots**
**of log-transformed $^{210}$Pb$_{ex}$ against mass depth (g cm$^{-2}$): (a) high-quality $^{137}$Cs and high-quality $^{210}$Pb (S-LO-I-W3-T1-CW-R1); (b)**
**high-quality $^{137}$Cs and low-quality $^{210}$Pb (M-OA-I-W4-T3-CW-R3).**

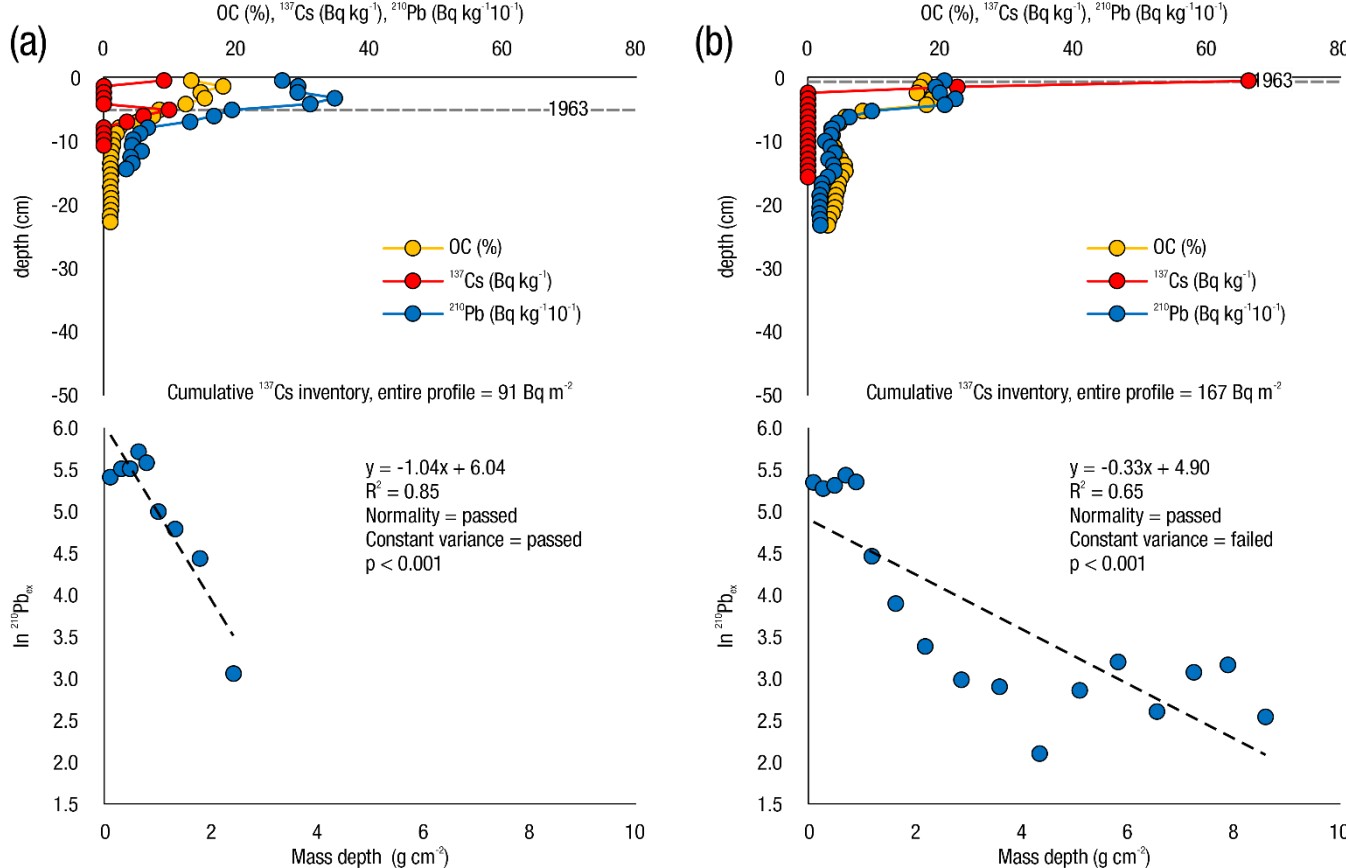

Figure 4: Examples of ¹³⁷Cs and ²¹⁰Pb classifications showing OC (%), ¹³⁷Cs (Bq kg⁻¹), and ²¹⁰Pb (Bq kg⁻¹) depth profiles and plots of log-transformed ²¹⁰Pb$_{ex}$ against mass depth (g cm⁻²): (a) low-quality ¹³⁷Cs and high-quality ²¹⁰Pb (S-RO-I-W1-T3-CW-R3); and (b) low-quality ¹³⁷Cs and low-quality ²¹⁰Pb (S-RO-I-W6-T1-CW-R1).

## 3.2    ¹³⁷Cs vs. ²¹⁰Pb derived organic carbon sequestration rates

For each of the four datasets (D1-D4), the points on the Q-Q plot were distributed in a straight line, showing a linear relationship between the two estimates being compared ($R^2 > 0.95$, p-value $< 0.001$) (Fig. 5).

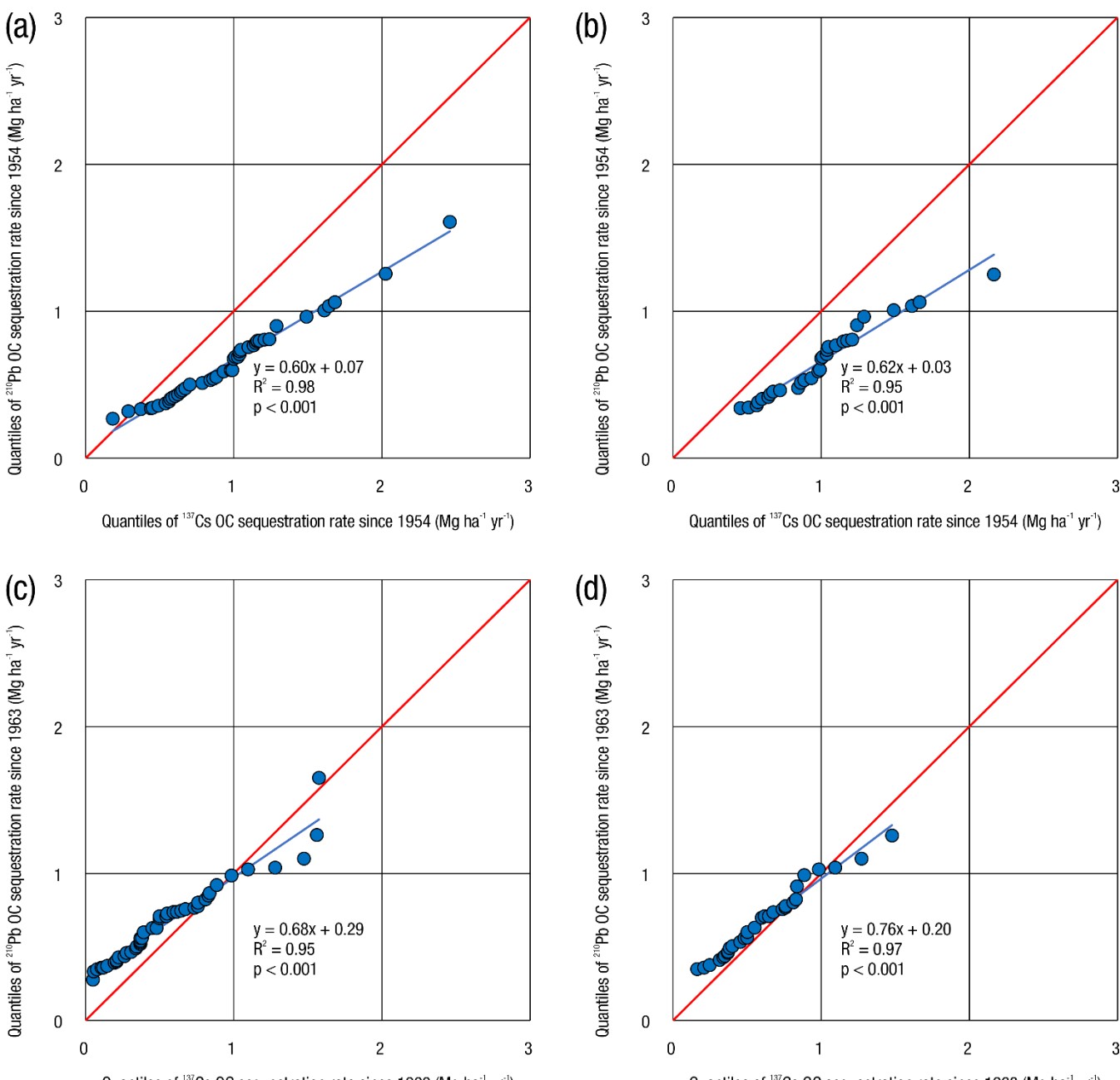

Figure 5: Q-Q plot of [137]Cs- vs. [210]Pb-based organic carbon (OC) sequestration rates using (a) all suitable [137]Cs and [210]Pb profiles estimated since 1954 (D1), (b) high-quality [137]Cs and [210]Pb profiles estimated since 1954 (D3), (c) all suitable [137]Cs and [210]Pb profiles estimated since 1963 (D2), and (d) high-quality [137]Cs and high-quality [210]Pb profiles estimated since 1963 (D4).

Visual inspection of the Q-Q plots showed that the points for D2 (i.e., all suitable $^{137}$Cs and $^{210}$Pb profiles using the 1963
time-marker; Fig. 5c) and D4 (i.e., high-quality $^{137}$Cs and $^{210}$Pb profiles using the 1963 time-marker; Fig. 5d) were
distributed more closely along the 1:1 line compared to that of D1 (i.e., all suitable $^{137}$Cs and $^{210}$Pb profiles using the 1954
time-marker; Fig. 5a) and D3 (i.e., high-quality $^{137}$Cs and $^{210}$Pb profiles using the 1954 time-marker; Fig. 5b).

An intercept closer to 0 and a slope closer to 1 indicated good alignment of the regression line to the 1:1 line. The slope (s)
and intercept (i) of the regression lines were: s = 0.60, i = 0.07 for D1 (Fig. 5a); s = 0.62, i = 0.03 for D3 (Fig. 5b); s = 0.68, i
= 0.29 for D2 (Fig. 5c); and s = 0.76, i = 0.20 for D4 (Fig. 5d). D2 and D4 had regression lines and slopes closer to the 1:1
line but intercepts further from the origin than D1 and D3.

The Cramer-von Mises test was used to build distance sampling models using the point-to-1:1-line distances computed from
the Q-Q plots. Models built with the D4 dataset produced the best-fit model (i.e., p-value > 0.05, AIC = -114). Models built
with the D1, D2, and D3 datasets had weaker p-values (p-value < 0.05) and can be ranked based on lower AIC scores (AIC =
-116 for D2, AIC = -54 for D1, and AIC = -34 for D3).
**3.3     Sediment accumulation, organic carbon sequestration rates and stocks**
The 30 sediment cores (68% of all the suitable $^{137}$Cs and $^{210}$Pb profiles) with high-quality $^{137}$Cs and $^{210}$Pb profiles were used
to calculate mass or sediment accumulation rates, OC sequestration rates, and OC stocks (Table 2). OC sequestration rates
based on $^{137}$Cs and $^{210}$Pb dating estimated since 1954 and 1963 of 44 suitable sediment cores (where both $^{137}$Cs and $^{210}$Pb
profiles were available) are presented in Supplementary Table 2.
**Table 2: Sedimentation accumulation, OC stocks, and sequestration rates of undisturbed wetlands estimated using high-quality**
**$^{137}$Cs and high-quality $^{210}$Pb profiles.**

| Type of radiometric dating | $^{137}$Cs | | $^{210}$Pb | |
|---|---|---|---|---|
| Time-marker | 1954 | 1963 | 1954 | 1963 |
| Range of accumulated sediment (Mg ha$^{-1}$) | 214-1,727 | 56-1,272 | 111-1,014 | 95-874 |
| Mean (standard deviation) stock of OC (Mg ha$^{-1}$) | 66 (29) | 35 (19) | 43 (18) | 38 (15) |
| Mean (standard deviation) rate of OC sequestration (Mg ha$^{-1}$ yr$^{-1}$) | 1.02 (0.44) | 0.63 (0.34) | 0.67 (0.27) | 0.68 (0.26) |


Based on the 1954 time-marker, the total sediment accumulation ranged from 214-1,727 Mg ha$^{-1}$ using $^{137}$Cs dating and 111-
1,014 Mg ha$^{-1}$ using $^{210}$Pb dating. In contrast, the total sediment accumulation based on the 1963 time-marker was lower,
ranging from 56-1272 Mg ha$^{-1}$ using $^{137}$Cs and 95-874 Mg ha$^{-1}$ using $^{210}$Pb dating.

The $^{137}$Cs-derived mean OC sequestration rate was almost two times larger, at 1.02 Mg ha-1 yr-1 using the 1954 time marker
compared to 0.63 Mg ha-1 yr-1 using the 1963 time marker. The corresponding $^{137}$Cs-based mean OC stocks were 66 Mg ha$^{-}$
$^{1}$ for 1954 and 35 Mg ha$^{-1}$ for 1963 (Table 2).

The $^{210}$Pb-derived mean OC sequestration rate was similar at 0.67 Mg ha$^{-1}$ yr$^{-1}$ using the 1954 time-marker compared to 0.68
Mg ha$^{-1}$ yr$^{-1}$ using the 1963 time-marker. $^{210}$Pb-based OC sequestration rates show minimal variation since they were
calculated using the same sedimentation rate. The corresponding mean OC stocks were 43 Mg ha$^{-1}$ for the 1954 time-marker
and 38 Mg ha$^{-1}$ for the 1963 time-marker, with a variable depth.

Figure 3 and Supplementary Figs. 1 to 12 present the depth distributions of $^{137}$Cs and $^{210}$Pb activity (along with the linear plot
of log-transformed $^{210}$Pb$_{ex}$ against mass depth in g cm$^{-2}$) of all suitable profiles (n = 44) where both radioisotope profiles are
available.
**4       Discussion**
This study compared $^{137}$Cs and $^{210}$Pb dating for OC estimates in wetlands that were undisturbed (i.e., without direct impact
human activities) since both radioisotopes dating are known to provide reliable forecasts for recent OC sequestration rates
(i.e., post-1954, which coincides with the onset of $^{137}$Cs atmospheric deposition) (Drexler et al., 2018; Creed et al., 2022).

This study highlights some advantages and disadvantages of using $^{137}$Cs vs. $^{210}$Pb dating. For example, the smaller number of
suitable $^{210}$Pb profiles (47/90 = 52%) due to the lack of a complete decay profile (following the CFCS model as described in
Sanchez-Cabeza and Ruiz-Fernandez, 2012) indicates that $^{210}$Pb dating is more prone to disturbance than $^{137}$Cs (79/90 =
88%). For $^{137}$Cs, even if the sediment core is disturbed, estimation of OC sequestration rates may be possible with careful
interpretation (e.g., see Fig. 2). The larger number of sediment cores using $^{137}$Cs dating can be beneficial in accurately
representing the heterogeneity of OC sequestration rates as it provides a larger dataset (a 36% gain compared to $^{210}$Pb).

Other advantages and disadvantages of $^{137}$Cs vs. $^{210}$Pb radioisotope dating are presented in Table 3. $^{137}$Cs deposition was a
pulse that occurred in 1954 and 1963. At the 1963 peak, the activity declined with time because of two factors: (1) peak

natural radioactive decay, with the $^{137}$Cs 30-year half-life reducing the peak size over time, and (2) peak attenuation due to physical, chemical, or biological reasons (Drexler et al. 2018). The declining $^{137}$Cs activity limits its applicability as a radioisotope dating tool; however, recent studies have reported adequate $^{137}$Cs reference inventories for Canadian landscapes (Sutherland, 1991; Kachanoski and Von Bertoldi, 1996; Li et al., 2008; Mabit et al., 2014; Zarrinabadi et al., 2023). In addition, the use of $^{137}$Cs inventory for dating to complement the peak has addressed the potential inadequacies that could be attributed to declining peak resolution with time. $^{137}$Cs dating is advantageous for its simplicity in pre- and post-processing of samples and the presence of additional time-markers in other regions (Breithaupt et al., 2018; Foucher et al., 2021). For example, additional time-markers correspond to the 1986 Chernobyl nuclear plant accident and 2011 Fukushima accident. However, their effect has yet to be recorded in North America due to the substantial distance from the source. Recognizing that there may be regional or local variation in peaks, we used non-eroded $^{137}$Cs reference sites to deal with regional variation in deposition. We also used multiple sampling sites within wetlands to assess local variation in deposition. Further, we looked for evidence from Chernobyl and Fukushima nuclear events in our data but found none (data not shown).

Further, we looked for evidence from Chernobyl and Fukushima nuclear events in our data but found none (data not shown). $^{137}$Cs dating is best suited for where the total OC is sequestered since a fixed time-marker (1954 onset or 1963 peak) or the average OC sequestration rate is needed. In contrast, the atmospheric deposition of $^{210}$Pb is continuous and, therefore, not limited in its applicability as a radioisotope dating tool. $^{210}$Pb dating is best suited for where variable OC sequestration rates are needed over a more extended period (earlier than 1954). $^{210}$Pb dating is advantageous because its calculations are based on multiple points associated with progressive OC sequestration rates derived using a constant rate of supply model — including the 1954 onset and 1963 peak of $^{137}$Cs activity—improving the precision of the OC sequestration rates. This precision enables estimating OC sequestration rates when wetlands are not undisturbed (history of drainage or at different ages since restoration) and undisturbed (no history of drainage).

**Table 3: The advantages and disadvantages of using $^{137}$Cs and unsupported $^{210}$Pb ($^{210}$Pb$_{ex}$) to estimate wetland organic carbon (OC) sequestration rates.**

| Method of radiometric dating | $^{137}$Cs | $^{210}$Pb$_{ex}$ |
|---|---|---|
| Advantages | • Calculations are based on single points representing the peak (1963) and onset (1954) of the fallout. There are additional time-markers for Europe (1986 due to the Chernobyl nuclear accident) and Japan (2011 due to Fukushima Daiichi nuclear accident).<br>• Sedimentation peak may still be evident allowing estimation of OC sequestration rate even if parts of the sediment core are disturbed.<br>• Sedimentation rate can be estimated using gamma detection, which is non-destructive, so sample can be re-analyzed or used for other analyses.<br>• Less sample preparation time for gamma analysis.<br>• After the $^{137}$Cs activity is measured, post-processing of data is less challenging. | • Calculations are based on multiple points as there is continuous atmospheric deposition.<br>• Sedimentation rate can be estimated using two reliable methods i.e., both alpha and gamma detection.<br>• Less sample preparation time for gamma analysis compared to alpha.<br>• Gamma analysis is non-destructive, so samples can be re-analyzed for other analyses compared to alpha.<br>• Can run multiple samples at a time on a single detector in alpha method. |

| Method of radiometric dating | | 137Cs | 210Pbex |
|---|---|---|---|
| Disadvantages | • | Risk of mixing of restored and drained states when estimating OC sequestration rates due to specificity of the 1954 and 1963 time-markers (e.g., if drained and restored after 1963). | • Requires full profile of $^{210}$Pb to do the calculations, if the sediment core disturbed then it cannot be used to estimate OC sequestration rates. |
| | • | Declining atmospheric deposition and declining inventory due to radioactive decay (i.e., with no more nuclear testing, atmospheric deposition only comes from recent accidental releases from Chernobyl and Fukushima). | • Sensitive to vertical mobilization of sediments, but not as much as $^{137}$Cs. <br> • The alpha method is destructive, and therefore the sample is not available for re-use or re-analysis. |
| | • | Sometimes the peak is not distinct. | • The alpha method requires extra precaution using hydrochloric acid for digesting, heating, spiking with $^{209}$Po tracer (i.e., analysts come in direct contact with radioactive material $^{209}$Po and hot acid). |
| | • | Can be estimated using only one reliable method i.e., Gamma detection. | |
| | • | Can run only one sample at a time on a single detector. | |
| | • | Sensitive to vertical mobilization of sediments. | • The alpha method takes more time per sample (i.e., overnight digest followed by at least 48 h on the alpha counter), and is more labor intensive i.e., digest, engraving coins, plating, transferring into ensemble, etc.). |
| | • | Sensitive to declining $^{137}$Cs inventory due to radioactive decay. | |
| | • | Sensitive to changes in redox potential. | • The alpha method requires more technical expertise for post processing of the data. |
| | • | More sensitive to biological and chemical activity compared to $^{210}$Pb (e.g., $^{137}$Cs can be taken up by plants instead of sodium or potassium, and $^{137}$Cs is soluble and therefore subject to mobility into solution then moving up and down the core. | • Uncertainty of $^{210}$Pbex results derived from gamma analysis can be higher than alpha. |

## 4.1 Challenges in interpreting the $^{137}$Cs peak

A potential weakness of $^{137}$Cs radioisotope dating arises from the challenges in interpreting the disturbed 1963 peak. The noise in the 1963 peak in wetlands on agricultural landscapes can be due to the redistribution of sediments since wetlands are susceptible to receiving a large mass of sediments resulting from various erosional processes due to their positioning within the landscape (Lobb et al., 2011; Zarrinabadi et al., 2023). Soil erosion resulting from wind, water, and tillage can lead to higher or lower $^{137}$Cs levels (Li et al., 2010; Foucher et al., 2021; Zarrinabadi et al., 2023) in wetlands in agricultural landscapes. If $^{137}$Cs enriched soil from the surrounding landscape gets deposited on top of the wetland's original soil layer, it can increase the $^{137}$Cs inventory value (Walling and Quine, 1991; Li et al., 2010). The magnitude of $^{137}$Cs enrichment depends on whether sediment comes from surface or sub-surface layers (Li et al., 2010; Lal, 2020). For example, if the wetland receives $^{137}$Cs enriched topsoil post-1963, the $^{137}$Cs inventory would be higher than the $^{137}$Cs depleted subsoil.

The screening of $^{137}$Cs profiles (Fig. 2a) considered the redistribution of sediments within the landscape. It demonstrated that the difficulty in disturbed $^{137}$C profile interpretation can be reduced by investigating the cumulative $^{137}$C inventory value. A cutoff cumulative $^{137}$Cs inventory value can help exclude questionable profiles. The range of $^{137}$Cs reference inventory values from previous erosion studies within the study area (e.g., Sutherland, 1991; Kachanoski and Von Bertoldi, 1996; Zarrinabadi et al., 2023) can help in establishing and setting the cutoff cumulative $^{137}$Cs inventory value. The mean $^{137}$Cs reference inventory values in the four provinces of Canada where our wetland sites are located were utilized in this instance. The mean $^{137}$Cs reference inventory value estimated to be 1,684 Bq m$^{-2}$ (coefficient of variation (CV) = 49%) for three AB wetland sites (53° N and 113° W) (Zarrinabadi et al. 2023), 989 Bq m$^{-2}$ (CV = 20%) for seven SK wetland sites (51° N and 107° w) (Sutherland, 1991), 1,008 Bq m$^{-2}$ (CV = 17.9%) for nine SK wetland sites (51° N and 104° W) (Sutherland, 1991), 1,430 Bq m$^{-2}$ (CV = 8.6%) for five MB wetland sites (50° N and 100° W) (Zarrinabadi et al. 2023), 1,447 Bq m$^{-2}$ (CV = 8.8%) for three ON wetland sites (43.3° N and 80.3° W) (Kachanoski and Von Bertoldi, 1996) and 1,534 Bq m$^{-2}$ (CV = 1.7%) for three ON wetland sites (45.6° N and 74.8° W) (Kachanoski and Von Bertoldi, 1996). The $^{137}$Cs reference inventory values were decay-corrected to 2021 for comparability. The cutoff cumulative $^{137}$Cs inventory value for this study was selected by checking the minimum $^{137}$Cs reference inventory value of the local region, i.e., 546 Bq m$^{-2}$ (using values reported in Sutherland, 1991; Kachanoski and Von Bertoldi, 1996; Zarrinabadi et al. 2023). Hence, any $^{137}$Cs inventory value less than 500 Bq m$^{-2}$ was considered questionable and low-quality. Additionally, > 75% of $^{137}$C profiles had a cumulative $^{137}$Cs inventory value of > 500 Bq m$^{-2}$, indicating that our wetland sites' $^{137}$Cs reference inventory value is most likely around 500 Bq m$^{-2}$.

Variations in the $^{137}$Cs peak types (e.g., distinct, broadened, fluctuating, etc.) and in $^{137}$Cs inventory values in this study
suggested that the $^{137}$Cs profiles were impacted by various regional erosional processes in the surrounding agricultural
landscape. Recent evidence suggests that there may be an outward movement of sediment and $^{137}$Cs from the center of the
wetlands to the riparian area (Zarrinabadi et al., 2023), suggesting that the base $^{137}$Cs inventory value observed in the center
of wetlands from atmospheric deposition in the 1950s-1960s could be less than that of the non-eroded reference $^{137}$Cs values
from the surrounding catchment. A $^{137}$C inventory of a sediment core can further help assign the $^{137}$Cs peak. For example, the
$^{137}$Cs peak was repositioned in disturbed sediment cores with higher $^{137}$Cs inventory, where the first discernable peak after
the sharp rise from the onset of $^{137}$Cs activity and exceeding or around the reference value was assumed to be the original
$^{137}$Cs peak. $^{239+240}$Pu isotopes, like $^{137}$Cs, are a product of nuclear testing and can be used to identify the peak of $^{137}$Cs. Future
research will use $^{239+240}$Pu to replace $^{137}$Cs as $^{137}$Cs levels diminish.

## 4.2      Challenges in interpreting $^{137}$Cs and $^{210}$Pb profiles

Mobilization of $^{137}$Cs and $^{210}$Pb in the sediment often occurs in wetlands. Vertical mixing of $^{137}$Cs within sediments can be
caused by remobilization and redistribution by wind and water, ice movement and inversion, disturbance by animals, and
disturbance by humans that ditch and drain the wetlands till through the wetland when it is dry and let cattle access them for
water which causes disturbances to the bottom sediments (Robbins et al., 1977; Milan et al., 1995; Takahashi et al., 2015).
Vertical mixing affects the profile by attenuating the peak upward and downward (which we addressed using the $^{137}$Cs
inventory value and not just the peak when assessing the profile). Horizontal mixing of $^{137}$Cs dating within sediment occurs
by physical movement of sediments into or out of the wetland, causing uneven distribution of the OC content, where
accumulation may be high at the edges of open water of the wetland (Lobb et al., 2011; Zarrinabadi et al., 2023). This
heterogeneity can be caused by the horizontal focusing of sediments in sub-basins within a wetland, i.e., multiple center
points. Sampling multiple sediment cores from individual wetlands can help capture the heterogeneity within the wetland.
Suppose the 137Cs activity of most of the sediment cores from a particular wetland is noisy with a higher inventory value
(e.g., $^{137}$Cs profile of S-LO-I-W4-T2-CW-R2 in Supplementary Fig. 2a). In that case, the impact by erosional processes can
be deduced with higher certainty. The higher observed inventory value could result from the movement of enriched material
via erosion/lateral flow to the center of the wetland, increasing the number of $^{137}$Cs. In this study, the assumption of no
substantial downward mixing of $^{137}$Cs was supported by (1) sampling three cores from each wetland, (2) assessing the
sharpness of the rise of the peaks (a sharp rise means negligible mixing), (3) examining the cumulative $^{137}$Cs inventory value
and validating against the known reference level, (4) classifying $^{137}$Cs profiles, and (5) corroborating with $^{210}$Pb dating.

**4.3** **$^{137}$Cs vs. $^{210}$Pb derived OC sequestration rates and stocks**

$^{137}$Cs radioisotope dating using the 1954 or 1963 time-markers gives reasonable estimates of OC sequestration rates as compared to $^{210}$Pb radioisotope dating. The $^{137}$Cs-$^{210}$Pb Q-Q plot of the 1963 OC sequestration rates is closer to the 1:1-line, suggesting compatibility between $^{137}$Cs- and $^{210}$Pb-based estimates (Fig. 5c and 5d). Conversely, the $^{137}$Cs-$^{210}$Pb Q-Q plot of the 1954 OC sequestration rates showed more deviation from the 1:1 line; $^{137}$Cs-based OC sequestration rates were more dispersed and were higher than the $^{210}$Pb-based OC sequestration rates (Fig. 5a and 5b). The mean OC sequestration rates in Table 2 further verify the comparability of OC sequestration rates using the 1963 time-marker (mean $^{137}$Cs OC sequestration rate is 0.63 Mg ha$^{-1}$ yr$^{-1}$ while mean $^{210}$Pb OC sequestration rate is 0.68 Mg ha$^{-1}$ yr$^{-1}$). The dispersion using the 1954 time-marker (mean $^{137}$Cs OC sequestration rate is 1.02 Mg ha$^{-1}$ yr$^{-1}$ while mean $^{210}$Pb OC sequestration rate is 0.67 Mg ha$^{-1}$ yr$^{-1}$). Providing better sequestration rate estimates has consequences for estimating OC stocks with an improved degree of accuracy, which may provide policymakers with better tools to make informed carbon management decisions supported with data.

To put our findings into practice and in the broader OC sequestration perspective, we consider a scenario where two independent studies were performed using $^{137}$Cs and $^{210}$Pb (with the CFCS model) at the exact locations. If the cores were not selected based on the criteria we used to choose high-quality profiles, then these two studies' OC sequestration rate estimates are likely to disagree. However, we know and have demonstrated through our findings that they are linearly dependent, and the equation of our linear regression lines may be used to transform one estimate to the other. Consequently, if the cores were selected based on our selection criteria, then one can expect the OC sequestration rate estimates to have similar values, which alleviates the interpretation challenges of having two different estimates from two independent studies. This observation may help with consistency when disagreements in estimates are observed. Another practical application of our findings may be in data augmentation. For example, if we have $^{210}$Pb data for a set of locations and $^{137}$Cs data for other locations, the linear regression equation could transform $^{210}$Pb data to augment $^{137}$Cs data, and vice versa. This can help data-driven modelling approaches, whereas larger datasets help achieve robust modelling tools. Similarly, because OC stocks can be derived from sequestration rates for specific years, estimates derived using one radioisotope can be used to estimate OC from a dataset derived from another estimate, further contributing to the augmentation of the corresponding OC stock data.

Based on the results of this study, we recommend (1) using high-quality $^{137}$Cs and $^{210}$Pb profiles to estimate OC sequestration rates, (2) interpreting 137Cs profiles from agricultural landscapes carefully from the perspective of redistribution of sediments, (3) using both $^{137}$Cs and $^{210}$Pb to compare and validate estimates if logistic approves. However, in case where one had to choose between $^{137}$Cs and $^{210}$Pb we recommend (1) For $^{137}$Cs: use 1963 time-markers to estimate OC sequestration rates (compared to 1954) since it is found to be most comparable with $^{210}$Pb dating techniques (CFCS model), (2) For $^{210}$Pb

(CFCS model): OC sequestration rates from present to 1963 can be estimated with highest precision since we corroborated
the estimates with $^{137}$Cs. However, we cannot comment on the precision of $^{210}$Pb-based OC sequestration rate estimation
before 1963 based on the scope of this study.
**5        Conclusions**
Information regarding OC sequestration rates within temperate inland wetland soils is crucial for evaluating the potential of
these ecosystems to serve as natural climate solutions. Radiometric dating using $^{137}$Cs and $^{210}$Pb presents a valuable tool for
estimating the recent OC sequestration potential of wetlands. Notably, a robust 1:1 linear correlation has been observed
between $^{137}$Cs- and $^{210}$Pb-based OC sequestration rates in high-quality sediment profiles.

While estimations based on the onset of $^{137}$Cs in 1954 or its peak in 1963 were reasonable, estimates anchored to the 1963
peak of $^{137}$Cs exhibited closer alignment with those derived from $^{210}$Pb data (using the CFCS model). These findings suggest
that estimates derived from $^{137}$Cs and $^{210}$Pb radioisotope dating methods are interchangeable and reasonably comparable
when utilizing the 1963 $^{137}$Cs time-marker.

Combining $^{137}$Cs and $^{210}$Pb tracers provides a comprehensive assessment of sedimentation rates. While one tracer offers an
average sedimentation rate over 60 years, the other provides a temporal trend over the same period. This interchangeability
enables more thorough evaluations of the average sedimentation rate in wetlands, which is crucial for leveraging them as
natural climate solutions.

**Code and data availability.** The R code for the distance sampling modelling along with the data to run the code is available at https://doi.org/10.5281/zenodo.10951658. The organic carbon (OC) sequestration rates data used to check the comparability of the radioisotope profiles can be found in the Supplement. These sequestration rate data and the geographical locations, years of sampling, and additional information about the sediment cores are available at https://doi.org/10.5281/zenodo.13696300. The radioisotope profiles used for screening are in the paper and Supplement. The paper and Supplement present other relevant data to support our conclusion.

**Author contributions.** The authors' contributions are as follows: PM: methodology, field and lab analysis, statistical analysis and modelling, writing; IFC: conceptualization, methodology, field and lab analysis, editing, supervision; CGT: conceptualization, editing, supervision; EE: methodology, field and lab analysis, editing; and DAL: methodology, field and lab analysis, editing.

**Competing interests**. The authors declare that they have no conflict of interest.

**Acknowledgements.** We acknowledge the support of the Natural Sciences and Engineering Research Council of Canada (NSERC): Strategic Partnership Grant (STPGP 506809) to IFC, DAL, and CGT. Additional funding sources are the NSERC Canadian Graduate Scholarship, Saskatchewan Innovation and Opportunity Scholarship, PhD Scholarship, School of Environment and Sustainability, University of Saskatchewan, and Wanda Young Scholarship awarded to PM. We thank Jacqueline Serran, Kevin Erratt, Oscar Senar, Ehsan Zarrinabadi, and many others for assisting in field sampling.

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
