# Peer review of "Technical Note: Comparison of radiometric techniques for estimating"

_EGUsphere, 2024_

## Author Response (AR1)

Note that the reviewer's original comments are in black and the author's response in blue.

**Response to comments of reviewer #1:**

The paper "Technical Note: Comparison of radiometric techniques for estimating recent organic carbon sequestration rates in freshwater mineral soil wetlands" by Mistry et al. evaluates the accuracy of an age model based on lead-210 and cesium-137 or the combination of both radionuclides for the reconstruction of organic carbon sequestration rates / stocks in Canada. In their paper, the authors evaluate the quality of the radionuclide profiles using a visual and statistical approach. On the selected profiles (classified as unperturbed), they compare the error associated with using the 1954 and 1963 time markers to establish age model in comparison to lead-210 age model for reconstructing the carbon sequestration rate.

The paper is interesting and well written, especially the discussion. The descriptions of the results were sometimes more complicated to follow. I have identified some points that should be clarified for the revised version. This paper is intended to be a 'technical note'. In my opinion, a technical note should be applicable worldwide. In the introduction, in Table 1 or again in the methods section, the authors mainly introduce 137Cs in North America (in the discussion they introduce Fukushima and Chernobyl peaks). This radionuclide has some specificities around the world that need to be introduced for a technical note. It will also be interesting to introduce and discuss other complementary approaches that can help to identify the 137Cs, the advantages, disadvantage of these techniques (e.g. Pu isotopes) or again the 'classic' combination of 137Cs and 210Pb to establish an age model.

**Authors' Response:**

The focus of our research is soil erosion and sedimentation within North America, which is only one region of the world. We have looked for evidence from Chernobyl and Fukushima $^{137}$Cs deposition events, but have not found any secondary peaks or elevated levels of activity in our research. Although there may be challenges applying our study to some parts of the world, we believe that the information is generally applicable and valuable for consideration in all regions. We would encourage others to further develop this approach for use in other regions where its application is not ideally suited. This point is also addressed in a few of our responses to the reviewer's comments below.

The focus of our research is on the inter-comparability of $^{137}$Cs and $^{210}$Pb in order to be able to bring data that are $^{137}$Cs or $^{210}$Pb together. The reviewer is correct that the ideal is to use $^{137}$Cs and $^{210}$Pb in combination. However, in the number of sediment cores collected in this study (90), the $^{137}$Cs profile was of high-quality at times but the $^{210}$Pb was not, and vice versa. In these cases, we wanted to explore the use of single tracer. Clearly, it is best to use both in tandem, to have $^{137}$Cs support the $^{210}$Pb dating.

The reviewer raises the potential of using Pu isotopes. Like $^{137}$Cs, Plutonium-239+240 ($^{239+240}$Pu) is a product of anthropogenic activities associated with nuclear bomb testing with a peak around 1963 similar to $^{137}$Cs. There are some advantages and disadvantages associated with $^{239+240}$Pu compared to $^{137}$Cs, summarized below.

**Advantages**

• $^{239+240}$Pu has a longer half life (24,110 years for $^{239}$Pu and 6,561 years for $^{240}$Pu) compared to $^{137}$Cs with a half-life of 30.2 years, hence approaching detection limits and potential obsolescence (Drexler et al., 2018). So $^{239+240}$Pu will continue to be reliably used for a much longer time unlike $^{137}$Cs.

- Less time consuming if $^{239+240}$Pu is analyzed using ICP-MS or Accelerator Mass Spectrometry (AMS) which takes minutes instead of hours that it takes to analyze $^{137}$Cs using gamma spectroscopy (Mabit et al., 2018).

- Smaller soil volume needed for analysis (Meusburger et al., 2023)

More relevant to Southern hemisphere location which received low initial $^{137}$Cs fallout (Tims et al., 2010) due to the longer half life of $^{239+240}$Pu compared to $^{137}$Cs

- Much more $^{239+240}$Pu was deposited compared to $^{137}$Cs (up to six times more), hence much more viable for measurements (i.e., more is always better for measurement).

- Background free, and therefore yielding results that are more precise than obtained for $^{137}$Cs (Fifield, 2008).

**Disadvantages of $^{239+240}$Pu compared to $^{137}$Cs**

- $^{239+240}$Pu is more suitable for field scale measurements while $^{137}$Cs is suitable for larger scales from plots to large watersheds, giving it larger applicability.

- While both $^{137}$Cs and $^{210}$Pb can be derived from gamma spectroscopy, $^{239+240}$Pu requires ICP-MS, alpha spectrometry or accelerator mass spectrometry, therefore there is no option of acquiring both $^{239+240}$Pu and $^{210}$Pb from a single run of sample (i.e., there will always be an additional cost and time if used to validate $^{210}$Pb).

We included in a revised manuscript, the following statement (**Lines 94-96**):

"Plutonium (Pu) may replace $^{137}$Cs in the future due to concerns of half-life and persistence as a dating tool. In essence, 239+240Pu has the same source and deposition mechanism as $^{137}$Cs. Its longer half-life will make its peak measurable when $^{137}$Cs is no longer measurable."

**Citations**

Drexler, J.Z., Fuller C.C., and Archfield S.: The approaching obsolescence of $^{137}$Cs dating of wetland soils in North America, Quaternary Science Reviews, 199: 83 – 96, doi:10.1016/j.quascirev.2018.08.028, 2018.

Mabit, L., Bernard, C., Yi, A.L.Z., Fulajtar, E., Dercon, G., Zaman, M., Toloza, A., Heng, L.: Promoting the use of isotopic techniques to combat soil erosion: An overview of the key role played by the SWMCN subprogramme of the Joint FAO/IAEA division over the last 20 years, Land Degradation and Development, VOLUME, 1-15. doi:10.1002/ldr.3016, 2018.

Meusburger, K., Porto, P., Waldis, J.K., and Alewell, C.: Validating plutonium-239+249 as a novel soil redistribution tracer – a comparison to measured sediment yield, Soil, 9: 399 – 409. doi:10.5194/soil-9-399-2023, 2023.

Fifield, L. K.: Accelerator mass spectrometry of the actinides, *Quaternary Geochronology*, *3*: 276-290. doi:10.1016/j.quageo.2007.10.003, 2008.

Tims, S.G., Everett S.E., Fifield L.K., Hancock G.J., and Bartley, R.: Plutonium as a tracer of soil and sediment movement in the Herbert River, Australia, Nuclear Instruments and Methods in Physics Research 268: 1150 – 1154. doi:10.1016/j.nimb.2009.10.121, 2010.

If I understand correctly, the authors compare the efficiency of both 137Cs age models (using 1954 and/or 1963 peaks) with the 210Pb model after the identification of 'high-quality profiles'. To me, these two approaches are complementary. Most studies using lake sediments, for example, validate their 210Pb age model by comparing the continuous lead chronology with the 137Cs peaks (1954 and 1963). The authors state that they use this combination, but this information is not clearly stated in the text (or I may have misunderstood them). Do they compare the validity of the 210Pb models in the conventional way (I don't think so) or simply by comparing the 137Cs and 210Pb profiles, which they think are of good quality?

**Authors' Response:**

We agree.  As noted in our responses to the reviewer's comments below, the $^{210}$Pb and $^{137}$Cs techniques are complementary, providing different methods of dating sediments.  $^{210}$Pb provides a more complex result, using a constant supply rate model to reveal trends in sedimentation, whereas $^{137}$Cs provides a simple result (i.e., an average sedimentation rate).  Together, they provide a more robust means of dating sediments.

However, the focus of our research was to explore the possibility of using them interchangeably. Not all cores provide high quality $^{137}$Cs and $^{210}$Pb data for the purposes of dating, so having multiple techniques increases the utility of sediment cores that are collected, often at great expense.  That is, if one of the two radioisotopes has a high-quality profile while the other one does not, either could be used to maximize the data of interest (in our case, organic carbon (OC) sequestration rate), instead of discarding a core because one of the radioisotopes is undesirable.

Having multiple techniques raises the obvious question, "Which is better?".  In our paper, we do not intend to identify which technique is better; we are attempting instead to compare the two techniques, highlighting the pros and cons of each, and assessing the degree to which they can be used interchangeably when one may not apply to a given core.  Also, as noted in our responses below, we don't view $^{137}$Cs data as a means to validate $^{210}$Pb results. Our perspective is that $^{210}$Pb and $^{137}$Cs provide two different measures of sedimentation rate—separate, but complementary—providing a better understanding of the sedimentation history.

At various points in the manuscript, the authors emphasize the simplicity of measuring 137Cs compared with 210Pb. This statement is true if researchers use alpha spectrometry, but as reported in some recent reviews (cited by the authors), studies using alpha spectrometry are decreasing, while gamma spectrometry is increasingly used. This technique makes it easy to obtain 137Cs, 210Pb and 241Am. At the end of the introduction, the authors state that "this study helps to reduce the uncertainty in studies that rely on 137Cs or 210Pb radioisotope dating". If you have access to both radionuclides with gamma spectrometry, you don't have to choose between the two because you have access to both radionuclides. The authors should justify why it is important to be able to run the lead-210 and 137Cs models separately.

**Authors' Response:**

A good question, why do we use both alpha and gamma spectroscopy when both $^{210}$Pb and $^{137}$Cs can be measured using gamma?  This is addressed in our responses to the comments below.  In brief, the quality of data for $^{210}$Pb using gamma spectroscopy is not as good

as $^{137}$Cs, and, the analytical requirements are substantially greater. Even with the best equipment and analytical procedures for measuring $^{210}$Pb with gamma, we often use alpha spectroscopy to confirm gamma measures and generate better data. In our laboratory we have many gamma and alpha spectrometers, so we have the luxury to use both.

The authors of this paper have done important field and analytical work that deserves to be published in an open access database. This will allow other researchers to evaluate their work and share these data with the community. I recommend that the authors make this dataset available for the revised version.

**Authors' Response:**

Thank you for this comment. We have provided a summary of our analyses in this manuscript (e.g., see **Supplementary Material**), and we are preparing an extensive data report for publication.

I recommend publishing this article after revision.

General comments:

Line 13: replace measuring with reconstructing?

**Authors' Response:**

Thank you for the suggestion. As we are measuring rates of sedimentation and OC accumulation in the sediments, we have opted to stay with the word "measuring".

Line 14: You mention 'a single point' when you actually mention two in the rest of the sentence (1954 and 1963). There may be other regional peaks (1986, 2011) or even more local ones.

**Authors' Response:**

It is true that there may be regional peaks, or even local variations in peaks. There are two time-markers or points-of-interest associated with $^{137}$Cs fallout: the onset and the peak. Atmospheric deposition has varied over time and around the globe. In the Americas, $^{137}$Cs deposition has been dominated by the thermonuclear activity between 1958 and 1963. This deposition has not been uniform, and regional and local variation is associated with variation in weather (wind and precipitation) during that period. In our studies in Central and South America, $^{137}$Cs levels are about one half of those in North America due to historical global weather patterns. We have also observed the local influence of rain shadowing by mountains. So, we agree that $^{137}$Cs deposition is regional and there is potential for regionally specific peaks. For this reason, we use non-eroded reference field sites to deal with regional variation in deposition, and we use multiple local sampling sites and repeated sampling (changes in values over time) to deal with local variation in deposition. We have looked for evidence from Chernobyl and Fukushima events but have not found any secondary peaks or elevated levels of activity in our research in the Americas.

Since Line 14 is in the abstract we added the following text to the Discussion section in the revised manuscript to clarify: (**Lines 597-600**)

"Recognizing that there may be regional or local variation in peaks, we used non-eroded $^{137}$Cs reference sites to deal with regional variation in deposition, and we used multiple sampling sites within wetlands to assess local variation in deposition. Further, we looked for evidence from Chernobyl and Fukushima nuclear events in our data but found none (data not shown)."

Line 19: You compare your 137Cs age models with the 210Pbxs age model? How?

**Authors' Response:**

It would be more accurate to state that $^{137}$Cs provides a measure of the average rate of sedimentation between the $^{137}$Cs time markers (1954 and 1963) and the date of sampling, whereas $^{210}$Pb$_{ex}$ uses a continuous rate of supply model to provide an estimate of sedimentation over time prior to the date of sampling. These rates of sedimentation, although very different in their means of determination, can be compared in the context of their periods of interest. In our study, we are interested in the rates of sedimentation over the past ~60 years.

We revised the text in the revised manuscript (**Lines 18-20**):

**Original:** "To this end, we propose a decision framework for screening $^{137}$Cs and $^{210}$Pb profiles into high- and low-quality sediment profiles, and we compare dating using the 1954 and 1963 time-markers."

**Revised:** "To this end, we propose a decision framework for screening $^{137}$Cs and $^{210}$Pb profiles into high- and low-quality sediment profiles, and we compare dating using the 1954 and 1963 time-markers, i.e., the rates of sedimentation and, consequently, OC sequestration over the past ~60 years."

Line 24: Perhaps a broader view here to introduce the importance of your topic? Area occupied by these wetlands in the world, Canada, etc.? Data about carbon sequestration by these wetlands?

**Authors' Response:**

Thank you for your suggestion.

We revised (and added) the following text in the revised manuscript (**Lines 23-46**):

**Original:** "Wetlands in agricultural landscapes serve a crucial role in providing habitat for wildlife, regulating climate, improving water quality, and preventing floods. Moreover, these wetlands have the potential to sequester organic carbon (OC), making them candidates to be natural climate solutions by offsetting carbon emissions. To estimate the OC sequestration potential, it is critical to establish precise measurements to quantify wetland OC sequestration, develop strategies to promote conservation and restoration efforts, incorporate carbon credits in the carbon markets, and validate the wetland-based ecosystem services."

**Revised:** "Wetlands in agricultural landscapes serve a crucial role in providing habitat for wildlife, regulating climate, improving water quality, and reducing floods. Moreover, these wetlands have the potential to sequester organic carbon (OC), making them candidates to be natural climate solutions by offsetting carbon emissions. These wetlands embedded in agricultural landscapes are recognized as temperate inland wetland soils. The global carbon stock of temperate inland wetland soils is estimated to be 46 Pg C to a

2 m depth, while Canada's temperate inland wetland soils are estimated to contain 4.6 Pg C (Bridgham et al., 2006.Compared to peatlands, the rapid rate of OC sequestration and the larger spatial extent of temperate inland wetland soils can help contribute significantly to regional or national carbon sequestration (Bridgham et al., 2006; Nahlik and Fennessey, 2016).

Canada encompasses around 25% of the world's wetlands, with an area of approximately 1.29 million square kilometers, which accounts for 13% of the country's terrestrial area (Environment and Climate Change Canada, 2016), highlighting the global importance of these wetlands. Unfortunately, there is extremely limited data on the OC sequestration rates in these wetlands. To estimate the OC sequestration potential of these wetlands, it is critical to establish precise measurements to quantify wetland OC sequestration, develop strategies to promote conservation and restoration efforts, incorporate carbon credits in the carbon markets, and validate the wetland-based ecosystem services."

**Citations:**

Bridgham, S. D., Megonigal, J. P., Keller, J. K., Bliss, N. B., and Trettin, C.: The carbon balance of North American wetlands, Wetlands, 26(4), 889-916, doi:10.1672/0277-5212(2006)26[889:TCBONA]2.0.CO;2, 2006.

Environment and Climate Change Canada (ECCC): Canadian Environmental Sustainability Indicators: Extent of Canada's Wetlands, ECCC, Gatineau, Quebec, www.ec.gc.ca/indicateurs-indicators/default.asp?lang=en&n=69E2D25B-1, last access: 10 July 2024, 2016.

Nahlik, A. M. and Fennessy, M. S.: Carbon storage in US wetlands. Nature Communications, 7(1), 1-9, doi:10.1038/ncomms13835, 2016.

Line 25: critical -> important?

**Authors' Response:**

We replaced the word "critical" by the word "important".

Line 40: formed -> produced during?

**Authors' Response:**

We replaced the world "formed" by the phrase "produced during".

Line 41: 1963 in the northern hemisphere. What about the southern hemisphere? If the article is to be a technical note applicable worldwide, it needs to give a more global view of 137Cs fallout (not just the 1963 peak).

**Authors' Response:**

We agree that it would be ideal if this technical note provided a more global view. Although there may be challenges applying our study to some parts of the world, we believe that the information is generally applicable and valuable for consideration in all regions. We encourage others to further develop this approach for use in other regions where its application is not ideally suited.

We revised and added the following text in the revised manuscript (**Lines 58-64**):

**Original:** "[137]Cs is an artificial radioisotope which was formed due to thermonuclear bomb testing in the 1950s and 1960s, with the onset of atmospheric deposition in 1954 and a peak in 1963 (Ritchie and McHenry, 1990)."

**Revised:** "[137]Cs is an artificial radioisotope that was produced during thermonuclear bomb testing in the 1950s and 1960s, with the onset of atmospheric deposition in 1954 and a peak in 1963 (Ritchie and McHenry, 1990). The testing caused radioactive uranium to decay, and, as a result, [137]Cs isotope was released into the atmosphere, which was then deposited around the globe. Although there may be challenges in applying our study to some parts of the world, the information is generally applicable and valuable for consideration in all regions. We encourage others to customize this approach further for use in other regions where Cs deposition histories vary."

Line 44: 1954 and 1963 - 1954 or 1963. Most publications using the 137Cs age model use both the 1954 and 1963 time points to construct their age models. When using two time points, the sedimentation rate is not always (often) constant.

**Authors' Response:**

It is true that when using two time points, the sedimentation rate is not always constant. Using the two time-markers for [137]Cs, we do not expect the sedimentation rates to be equal as there may be differences associated with the processes operating between 1954 and 1963. However, we do expect the sedimentation rates to be similar. This redundancy in measurement is largely a means to deal with the uncertainties in identifying either time-marker alone.

We also recognize that the sedimentation rate between sampling and 1954 and 1963 will not likely be constant. Much of the sedimentation that occurs in these wetlands is impacted by the land use and land management practices in the catchments that surround them, and that land use and land management has changed significantly over that 60-year time period. This is why we couple [137]Cs with [210]Pb techniques.

We revised and added the following text in the revised manuscript (**Lines 67-68**):

**Original:** "[137]Cs dating assumes constant sedimentation rates measured since 1954 and 1963."

**Revised:** "[137]Cs dating assumes constant sedimentation rates measured since 1954 or 1963. In using the two time-markers for [137]Cs, we do not expect the sedimentation rates to be equal but we do expect them to be similar."

Lines 47-48: In Europe the story is more complex and it is sometimes difficult to distinguish the 1963 peak from the 1986 peak. Authors should introduce additional markers (241Am, Pu isotopes) to help distinguish these peaks and avoid errors in age modeling.

**Authors' Response:**

We have the capacity to analyze [239+240]Pu with alpha spectroscopy, and we are moving to that isotope as a replacement for [137]Cs as [137]Cs levels diminish (30.18-year half-life versus 6,000+ year half-life). In the Americas, we do not see evidence of the 1986 [137]Cs peak which is observed in Europe, so we have not used Pu to distinguish the 1986 [137]Cs peak from the 1963 peak.

We added the following text in the revised manuscript (**Lines 84-88**):

**Original:** "$^{137}$Cs has an additional time-marker for Europe in 1986 due to the Chernobyl nuclear accident and for Japan in 2011 due to the Fukushima Daiichi nuclear accident (Foucher et al., 2021), indicating that OC sequestration estimates can be derived for different timescales."

**Revised:** "$^{137}$Cs has an additional time-marker for Europe in 1986 due to the Chernobyl nuclear accident and for Japan in 2011 due to the Fukushima Daiichi nuclear accident (Foucher et al., 2021), indicating that OC sequestration estimates can be derived for different timescales. In the Americas, we do not see evidence of the 1986 or 2011 $^{137}$Cs peaks which are observed in Europe and Japan, respectively, so we did not need to use other radioisotope techniques (e.g., $^{239+240}$Pu) to distinguish the 1986 or 2011 $^{137}$Cs peak from the 1963 $^{137}$Cs peak."

Line 54: I don't agree that the 210Pb models are "complicated", as the authors have already stated in the abstract. Lead models provide complementary information than 137Cs age models. There are many R codes that facilitate the establishment of 210Pb models (e.g. SERAC model, Bruel and Sabatier, 2020).

**Authors' Response:**

We agree that the $^{210}$Pb and $^{137}$Cs techniques are complementary, providing different methods of dating sediments. $^{210}$Pb provides a more complex result, using a supply rate model to reveal trends in sedimentation, whereas $^{137}$Cs provides a simple result, an average sedimentation rate. Neither method is complicated.

We clarified the text in the revised manuscript (**Lines 93-94**):

**Original:** "$^{137}$C dating calculations are less complicated than $^{210}$Pb, with little modelling knowledge or expertise needed (Breithaupt et al., 2018)."

**Revised:** "$^{137}$Cs dating provides a simple result (an average sedimentation rate), while $^{210}$Pb dating provides a more complex result (using a supply rate model to reveal trends in sedimentation rates)."

Line 64: remove 'for'

**Authors' Response:**

The word "for" was removed in the revised manuscript.

Line 67: or mass accumulation rate?

**Authors' Response:**

"Sediment accumulation rate" was replaced with "mass accumulation rate".

Line 70: only cite Appleby and Odfield?

**Authors' Response:**

We remove the other citations, citing only Appleby and Oldfield (1978) as suggested.

Line 71: In my opinion, 137Cs provide temporal markers as mentioned by the author, but not 210Pbex, which provides a continuous age model. The 137Cs help to validate the 210Pb model.

**Authors' Response:**

We agree.

We clarified by adding the following text to the revised manuscript (**Lines 129-133**):

**Original:** "Both $^{137}$Cs and $^{210}$Pb provide suitable time-markers and a longer time horizon compared to direct measurements using the time-marker of horizons (2-10 years) to study sediment accretion and, subsequently, OC sequestration rates in wetlands (Bernal and Mitsch, 2013; Villa and Bernal, 2018)."

**Revised:** "Both $^{137}$Cs and $^{210}$Pb provide suitable time-markers and a longer time horizon compared to direct measurements using the time-marker of horizons (2-10 years) to study sediment accretion and, subsequently, OC sequestration rates in wetlands (Bernal and Mitsch, 2013; Villa and Bernal, 2018). In this study, we compared the average OC sequestration rate derived from $^{137}$Cs temporal markers with the progressive OC sequestration rates derived using a constant rate of supply model applied to $^{210}$Pb."

Line 77: If you do gamma spectrometry, you get both210Pbex, which provides a continuous age model. The 137Cs help to validate the 210Pb model.

**Authors' Response:**

We agree. See next response.

Line 77: If you perform gamma spectrometry you obtain both 137Cs and 210Pb (and 241Am) and you don't have any extra cost, time and specialized equipment.

**Authors' Response:**

Using the most common gamma spectrometers (Coaxial Ge—old technology), there is greater uncertainty (smaller peak resolution) in the low energy region of the spectrum (< 100 keV) where $^{210}$Pb is measured, relative to $^{137}$Cs (662 keV). In our laboratory, we use more recent technology, BEGe gamma spectrometers, which provide a broader range of high efficiency across the spectrum, allowing reasonably accurate measurement of both $^{137}$Cs and $^{210}$Pb.

Even with the newer technology, we find that performing gamma spectrometry on $^{210}$Pb takes extra time and cost. The prepared samples must be sealed for weeks before analysis of $^{210}$Pb by gamma spectrometry; these sealed samples are necessarily small and, therefore, take longer to analyze (24-36 hrs for $^{210}$Pb vs. 12-16 hrs for $^{137}$Cs), which represents additional cost. In contrast, when we analyze for $^{137}$Cs only, samples can be large and can be analyzed relatively quickly.

Further, in our experience, there is much greater uncertainty in $^{210}$Pb results than $^{137}$Cs results using gamma spectrometry, and we often use alpha spectroscopy to corroborate $^{210}$Pb results.

In this manuscript, we don't view $^{137}$Cs data as a means to validate $^{210}$Pb results. Our perspective is that $^{210}$Pb and $^{137}$Cs provide two different measures of sedimentation rate—separate, but complementary—providing a better understanding of the sedimentation history.

Line 85: change intact through undisturbed?

**Authors' Response:**

We changed "intact" to "undisturbed".

Lines 86-87: methods?

**Authors' Response:**

We moved lines 86-87 to the "Methods" in the revised manuscript.

Line 94: This figure should be in the main manuscript.

**Authors' Response:**

We moved the map (Supplementary Figure 1) to the main manuscript.

Line 95: change 'intact' by undisturbed ?

**Authors' Response:**

We change "intact" to "undisturbed.

Lines 114-119: the authors must discuss what can change the shape of these peaks (e.g. bioturbation, erosion phenomena, deposition of 137Cs-labelled particles, climatic events, ...), see existing bibliographic reviews.

**Authors' Response:**

We added the following text in the revised manuscript (**Lines 241-264**):

"The magnitude and shape of the $^{137}$Cs peaks observed in the sediments can be affected by the atmospheric deposition rate of $^{137}$Cs, which is obviously affected by the number and magnitude of emission events and the weather conditions following these events (UNSCEAR, 2000). The magnitude and shape of these peaks are also impacted by the movement of water and sediment within each wetland's catchment during the peaks' development (Milan et al., 1995; Zarrinabadi et al., 2023). Here, changes in the shape of the peaks are caused by the upward and downward movement of the sediment within the sediment profile (the movement of $^{137}$Cs through

diffusion (Klaminder et al., 2012) is presumed negligible). Bioturbation can cause an upward and downward mixing of the $^{137}$Cs in the profile, resulting in peak attenuation (Robbins et al., 1977). Even wave action during the period of atmospheric deposition will have a similar attenuation effect (Andersen et al., 2000; Zarrinabadi et al., 2023). Following peak atmospheric deposition, soil erosion and the accumulation of sediment will deliver sediments to the top of the profile, and those sediments may be higher or lower in concentration depending on the degree of preferential sediment transport and the associated enrichment or depletion of $^{137}$Cs in the added sediment (Zarrinabadi et al., 2023)."

**Citations:**

Andersen, T. J., Mikkelsen, O. A., Møller, A. L., and Pejrup, M.: Deposition and mixing depths on some European intertidal mudflats based on $^{210}$Pb and $^{137}$Cs activities, Continental Shelf Research, 20(12-13), 1569-1591, doi: 10.1016/S0278-4343(00)00038-8, 2000.

Milan, C. S., Swenson, E. M., Turner, R. E., and Lee, J. M.: Assessment of the method for estimating sediment accumulation rates: Louisiana salt marshes, Journal of Coastal Research, 296-307, https://www.jstor.org/stable/4298341, 1995.

Robbins, J.A., Krezoski, J.R. and Mozley, S.C.: Radioactivity in sediments of the Great Lakes: post-depositional redistribution by deposit-feeding organisms, Earth Planet. Sci. Lett., 36, 325–333, doi: 10.1016/0012-821X(77)90217-5, 1977.

Klaminder, J., Appleby, P., Crook, P., and Renberg, I.: Post-deposition diffusion of $^{137}$Cs in lake sediment: Implications for radiocaesium dating, Sedimentology, 59: 2259–2267, doi:10.1111/j.1365-3091.2012.01343.x, 2012.

United Nations Scientific Committee on the Effects of Atomic Radiation (UNSCEAR), 2000. Sources and Effects of Ionizing Radiation, V1, United Nations, New York, doi:10.18356/49c437f9-en, 2000.

Zarrinabadi, E., Lobb, D. A., Enanga, E., Badiou, P., and Creed, I. F.: Agricultural activities lead to sediment infilling of wetlandscapes in the Canadian Prairies: Assessment of soil erosion and sedimentation fluxes, Geoderma, 436, 116525, doi:10.1016/j.geoderma.2023.116525, 2023.

Line 140: The authors should describe where these reference samples were collected and describe the environment (e.g. undisturbed grassland?). Authors use one reference core per site? This part was not clear for me

**Authors' Response:**

Thank you for this suggestion. The details of the number of reference sites used to generate reference $^{137}$Cs estimates are provided below which can also be found in the corresponding citation.

For the 3 wetlands in AB (53° N and 113° W)—2 reference sites used (a total of 24 sediment cores) to generate reference $^{137}$Cs estimates (Zarrinabadi et al., 2023).

For the 7 wetlands in SK (51° N and 107° W)—1 reference site used (a total of 3 sediment cores dividing into 20 samples) to generate reference $^{137}$Cs estimates (Sutherland, 1991).

For the 9 wetlands in SK (51° N and 104° W)—1 reference site used (a total of 3 sediment cores dividing into 20 samples) to generate reference $^{137}$Cs estimates (Sutherland, 1991).

For the 5 wetlands in MB (50° N and 100° W)—3 reference sites used (a total of 27 sediment cores) to generate reference $^{137}$Cs estimates (Zarrinabadi et al., 2023).

For the 3 wetlands in ON (43.3° N and 80.3° W)—1 reference site used (a total of 18 sediment cores compositing into 2 samples) to generate reference $^{137}$Cs estimates (Kachanoski and Von Bertoldi, 1996).

For the 3 wetlands in ON (45.6° N and 74.8° W)—1 reference site used (a total of 18 sediment cores compositing into 2 samples) to generate reference $^{137}$Cs estimates (Kachanoski and Von Bertoldi, 1996).

We have been successful in using catchment budgeting as an alternative method of establishing reference $^{137}$Cs values for specific wetlands, but this is far more time consuming and expensive than the standard approach.

We added the following text in the revised manuscript where we describe the environment and estimated location (from our wetland sites) of the reference sites (**Lines 289-293**):

"Ideally, reference sites are large, open, level, non-eroded areas, usually in forage or grassland since the 1950s, and within 10 km of the site of interest. In this study, it was impossible to identify a suitable reference site near every wetland; it is usually difficult to find reference sites in agricultural landscapes. However, we could locate reference sites used in other studies within 50 km except from nine wetlands in SK (51° N and 104° W), which were ~150 km from the reference site. Although this was not considered ideal, it was considered acceptable."

We also revised the $^{137}$Cs reference inventory values the in the Discussion section in the revised manuscript to present $^{137}$Cs reference inventory values near our wetland sites.  As mentioned earlier, details on the reference sites can be found in the corresponding citation (**Lines 642-648**).

"The mean $^{137}$Cs reference inventory values in the four provinces of Canada where our wetland sites are located were utilized in this instance. The mean $^{137}$Cs reference inventory value estimated to be 1,684 Bq m-2 (coefficient of variation (CV) = 49%) for three AB wetland sites (53° N and 113° W) (Zarrinabadi et al. 2023), 989 Bq m-2 (CV = 20%) for seven SK wetland sites (51° N and 107° W) (Sutherland, 1991), 1,008 Bq m-2 (CV = 17.9%) for nine SK wetland sites (51° N and 104° W) (Sutherland, 1991), 1,430 Bq m-2 (CV = 8.6%) for five MB wetland sites (50° N and 100° W) (Zarrinabadi et al. 2023), 1,447 Bq m-2 (CV = 8.8%) for three ON wetland sites (43.3° N and 80.3° W) (Kachanoski and Von Bertoldi, 1996) and 1,534 Bq m-2 (CV = 1.7%) for three ON wetland sites (45.6° N and 74.8° W) (Kachanoski and Von Bertoldi, 1996)."

**Citations:**

Kachanoski, R. G. and Von Bertoldi, P.: Monitoring soil loss and redistribution using $^{137}$Cs, COESA Report No. RES/MON-008/96, Green Plan Research Sub-program, Agriculture and Agri-food Canada, London, Ontario, 1996.

Sutherland, R. A.: Examination of caesium-137 areal activities in control (uneroded) locations, Soil technology, 4(1), 33-50, doi:10.1016/0933-3630(91)90038-O, 1991.

Zarrinabadi, E., Lobb, D. A., Enanga, E., Badiou, P., and Creed, I. F.: Agricultural activities lead to sediment infilling of wetlandscapes in the Canadian Prairies: Assessment of soil erosion and sedimentation fluxes, Geoderma, 436, 116525, doi:10.1016/j.geoderma.2023.116525, 2023.

Line 164: Use organic carbon instead of OC in the title.

**Authors' Response:**

We replaced "OC" with "organic carbon" in the title.

Line 169: the authors should explain how they get from g.cm2.year to Mg.ha.year-1 (representativeness of the core to extrapolate the value to the ha scale?)

**Authors' Response:**

Here, we multiplied g cm$^{-2}$ yr$^{-1}$ by 100 to get to Mg ha$^{-1}$ yr$^{-1}$.

We reported the data in Mg ha$^{-1}$ yr$^{-1}$ since it is widely used standardized unit to report OC sequestration rate data allowing comparability with other studies.

We add the following text in the revised manuscript (**Lines 351-353**):

"Unit conversion is applied to report the OC sequestration rate estimates in Mg ha$^{-1}$ yr$^{-1}$ from g cm$^{-2}$ yr$^{-1}$ for easy standardization and comparability with other studies."

Line 171 - 173: why do the authors not use the 241Am to more clearly identify the 1963 peak (when the shape of the peak is not clear)? If they are not sure about the peak, why not use the onset of 137Cs in 1954 instead of remove the profile from their selection (maybe I don't have understand here)?

**Authors' Response:**

We recognize the potential for the use of $^{241}$Am as an alternative to $^{137}$Cs, but we are moving towards $^{239+240}$Pu as an alternative. $^{241}$Am has similar challenges as $^{210}$Pb measured using gamma spectrometry: $^{241}$Am is detected in the low-energy range of the spectrum where detection is less accurate; i.e., it is more difficult to distinguish the energy peak from the background noise in the spectrum. We are not confident that $^{241}$Am would provide any greater ability to distinguish 1963 peaks in the sediment profile. The problem we are observing is the presence of wide peaks or multiple peaks spanning multiple sample increments, and this problem will exist with either $^{241}$Am or $^{137}$Cs.

Line 178: if you use density correction to build your age models, you should describe it in this section.

**Authors' Response:**

Yes, we used a density correction to build our continuous age models. The continuous age models were based on mass depth (g cm$^{-2}$) which is a function of sediment density, and not depth (cm).

We clarified the sentence by revising the following sentence (**Lines 359-361**).

**Original:** "The CFCS model uses the log-linear relationship of $^{210}Pb_{ex}$ with depth and converts $^{210}Pb_{ex}$ to the sediment accumulation rate and, consequently, the OC sequestration rate."

**Revised:** "The CFCS model uses the log-linear relationship of $^{210}Pb_{ex}$ with mass depth and converts $^{210}Pb_{ex}$ to the mass or sediment accumulation rate and, consequently, the OC sequestration rate."

Line 230: repositioned? what does that mean?

**Authors' Response:**

For some profiles, the highest activity of $^{137}Cs$ in the sediment profile was not always used as the basis for estimating the OC sequestration rate since 1963. Some sediment profiles can have very high total quantities of $^{137}Cs$, or profile inventories, with some of the post-1963 deposition of $^{137}Cs$ received in $^{137}Cs$-enriched sediments from the surrounding catchment. Sediments that have undergone substantial preferential detachment and entrainment on their pathway into a wetland can have very high concentrations of $^{137}Cs$, and when interlayered with sediments that are not so enriched can generate multiple $^{137}Cs$ peaks in the sediment profile peaks after 1963. The high $^{137}Cs$ profile inventory values along with interlayers of sediment would suggest this phenomenon. In such "noisy" profiles associated with higher $^{137}Cs$ inventories, the first discernible peak after the sharp rise from the onset of $^{137}Cs$ activity and exceeding or around the reference value was assumed to be the original $^{137}Cs$ peak.

To be clear, the multiple $^{137}Cs$ peaks and elevated inventories referred to here are not presumed to be associated with Chernobyl and Fukushima events. When multiple peaks are observed, they are local and not regional in nature, ruling out the global impacts of Chernobyl and Fukushima.

In the revised manuscript, clarification can be found in the Discussion section and in the revised text of the next comment.

Line 231: How do the authors explain the high stocks of these cores? Are these cores in an accumulation area?

**Authors' Response:**

The high stocks could, partly, be attributed to the location of the wetlands from which the cores were collected. The wetlands were situated with agricultural landscapes, where there was a large potential for high organic matter input from the surrounding landscape (i.e., allochthonous organic matter inputs), and due to nutrient-rich runoff from the surrounding landscape to the wetland, there was also a large potential for high organic matter generation in situ (i.e., autochthonous organic matter inputs), with consequences on OC stocks skewed towards larger values.

We added following text in the revised manuscript (Lines **435-445**):

**Original:** "Of the 62 high-quality $^{137}$Cs profiles, 4 (6.5%) were repositioned to capture the $^{137}$Cs enriched sediments post 1963. In these profiles, which had a cumulative $^{137}$Cs inventory value > 1,200 Bq m$^{-2}$, the depth that corresponded to $^{137}$Cs cumulative inventory value of ~500 Bq m$^{-2}$ was considered as the 1963 time-marker."

**Revised:** "Of the 79 suitable $^{137}$Cs profiles, 62 (78%) were classified as high-quality. Of the 62 high-quality $^{137}$Cs profiles, 61% had clear and distinct peaks, with a smooth rise and decline. In contrast, the remaining 39% had noise—either one-sided peaks or disturbed peaks (e.g., Fig. 4). Of the 62 high-quality $^{137}$Cs profiles, 4 (6.5%) were repositioned to capture the $^{137}$Cs enriched sediments post 1963 (e.g., $^{137}$Cs profile of S-LO-I-W4-T2-CW-R2 in Supplementary Fig. 2a). In these profiles, which had a cumulative $^{137}$Cs inventory value > 1,200 Bq m-2, the depth that corresponded to $^{137}$Cs cumulative inventory value of ~500 Bq m-2 was considered as the 1963 time-marker. The high total quantities of $^{137}$Cs profile inventories can be attributed to receiving $^{137}$Cs enriched sediments from the surrounding landscape. Sediments that have undergone substantial preferential detachment and entrainment on their pathway into a wetland can have very high concentrations of $^{137}$Cs and, when interlayered with sediments that are not so enriched, can generate multiple $^{137}$Cs peaks in the sediment profile peaks after 1963. These observed multiple peaks are local and not regional, ruling out the association with Chernobyl and Fukushima events. "

Lines 257-259: This part corresponds to the methodology already described in the Materials and Methods section. I will delete it from the results

**Authors' Response:**

We deleted the following sentence from the results in the revised manuscript.

**Original:** "The comparability of $^{137}$Cs vs. $^{210}$Pb derived OC sequestration rates was investigated through both visual inspection of the Q-Q plots and the Cramer-von Mises test which assigned significance of the distance of the points from the 1:1 line assessed with p-value and the AIC."

**Revised:** *TEXT DELETED*

Line 310: To the best of my knowledge, you need a complete decay profile to apply the CRS model, it's the same for the CFCS model? Can you give a reference for this statement?

**Authors' Response:**

Yes, we did achieve a complete decay profile as described in Sanchez-Cabeza and Ruiz-Fernandez (2012).

We clarified this in the revised manuscript (**Lines 562-565**):

**Original:** "For example, the significantly smaller number of suitable $^{210}$Pb profiles (47/90 = 52%) due to lack of a complete decay profile indicates that $^{210}$Pb dating is more prone to disturbance than $^{137}$Cs (79/90 = 88%).

**Revised:** "For example, the smaller number of suitable $^{210}$Pb profiles (47/90 = 52%) due to the lack of a complete decay profile (following the CFCS model as described in Sanchez-Cabeza and Ruiz-Fernandez, 2012) indicates that $^{210}$Pb dating is more prone to disturbance than $^{137}$Cs (79/90 = 88%)."

Line 311: I didn't understand if you corrected your 210Pb age model with density to avoid perturbations/inconsistencies/soil properties? please clarify.

**Authors' Response:**

Yes, we corrected our $^{210}$Pb age model with density to avoid perturbations. We used the methods described in Sanchez-Cabeza and Ruiz-Fernandez (2012) for the $^{210}$Pb age model and used mass depth which controls for compaction unlike the depth-based approach that would respond to compaction and affect the dates. We used mass depth (g cm$^{-2}$) because it does not respond to compaction and by extension bulk density as described by Sanchez-Cabeza and Ruiz-Fernandez (2012) and not the depth based (cm) which would be affected by changes in bulk density with consequences on depth.

We clarified this in the revised manuscript and added the following text to the Methods (**Lines 562-567**):

**Original:** "For example, the significantly smaller number of suitable $^{210}$Pb profiles (47/90 = 52%) due to lack of a complete decay profile indicates that $^{210}$Pb dating is more prone to disturbance than $^{137}$Cs (79/90 = 88%)."

**Revised:** "For example, the smaller number of suitable $^{210}$Pb profiles (47/90 = 52%) due to the lack of a complete decay profile (following the CFCS model as described in Sanchez-Cabeza and Ruiz-Fernandez, 2012) indicates that $^{210}$Pb dating is more prone to disturbance than $^{137}$Cs (79/90 = 88%). For $^{137}$Cs, even if the sediment core is disturbed, estimation of OC sequestration rates may be possible with careful interpretation (e.g., see Fig. 2). The larger number of sediment cores using $^{137}$Cs dating can be beneficial in accurately representing the heterogeneity of OC sequestration rates as it provides a larger dataset (a 36% gain compared to $^{210}$Pb)."

Line 326: Fukushima and Chernobyl releases are not 'less', just not recorded in North American lakes or wetlands to the best of my knowledge.

**Authors' Response:**

We agree. We revised the text to reflect that Fukushima and Chernobyl releases are not recorded in North American lakes (**Lines 595-597**).

**Original:** "For example, there are additional time-markers corresponding to the 1986 Chernobyl nuclear plant accident and 2011 Fukushima accident, although their effect is less felt in North America due to the substantial distance from the source."

**Revised:** "For example, additional time-markers correspond to the 1986 Chernobyl nuclear plant accident and 2011 Fukushima accident. However, their effect has yet to be recorded in North America due to the substantial distance from the source."

Line 331: continuous age model instead of 'can provide multiple time markers?

**Authors' Response:**

We replaced "can provide multiple time-markers" with the following text. Please note the previously provided explanation. We compared the average OC sequestration rate derived from $^{137}$Cs temporal markers with the progressive OC sequestration rates derived using a constant rate of supply model applied to $^{210}$Pb (**Lines 606-608**).

**Original:** "$^{210}$Pb dating is advantageous because its calculations are based on multiple points and can provide several time-markers—including the 1954 onset and 1963 peak of $^{137}$Cs activity—improving the precision of the OC sequestration rates."

**Revised:** "$^{210}$Pb dating is advantageous because its calculations are based on multiple points associated with progressive OC sequestration rates derived using a constant rate of supply model—including the 1954 onset and 1963 peak of $^{137}$Cs activity—improving the precision of the OC sequestration rates."

Line 345 : references for this statement?

**Authors' Response:**

We added the references Li et al. (2010) and Lal (2020) to support this statement:

Li et al. (2010) was already included in the manuscript.

Lal (2020) is now included in the manuscript.

Lal, R. (2020). Soil erosion and gaseous emissions. Applied Sciences, 10, 2784, doi:10.3390/app10082784, 2020.

Line 361: It was not possible to collect a reference site near each wetland?

**Authors' Response:**

As noted above, it was not possible to identify a suitable reference site (large, open, level, non-eroded area with 10 km) near each wetland.  It is extremely difficult to find such a site in agricultural landscapes.  We have been successful in using catchment budgeting as an alternative method of establishing reference $^{137}$Cs values for specific wetlands, but that is far more time consuming and expensive than the standard approach.

In section 4.1, the authors should discuss alternative methods that may help to identify the peak of 137Cs (e.g. Pu isotopes).

**Authors' Response:**

 As noted above, we have the capacity to analyze $^{239+240}$Pu with alpha spectroscopy, and we are moving to Pu as a replacement for $^{137}$Cs as $^{137}$Cs levels diminish.  We are into our third half-life for $^{137}$Cs (30.18-year half-life versus 6,000+ year half-life).

We added the following text in the revised manuscript (**Lines 712-713**):

"$^{239+240}$Pu isotopes which, like $^{137}$Cs, are a product of nuclear testing can be used to identify the peak of $^{137}$Cs. Future research will move to using $^{239+240}$Pu as a replacement for $^{137}$Cs as $^{137}$Cs levels diminish."

Lines 383-386: very long sentence...

**Authors' Response:**

We simplified the text in the revised manuscript (**Lines 726-729**).

**Original:** "If the $^{137}$Cs activity of most of the sediment cores from an individual wetland are noisy with a higher inventory value, then the impact by erosional processes can be deduced with higher certainty because the higher observed inventory value could be a result of movement of enriched material to the center of the wetland, therefore increasing the quantity of $^{137}$Cs from the value that would be expected if no new enriched material was introduced via erosion/lateral flow."

**Revised:** "Suppose the $^{137}$Cs activity of most of the sediment cores from a particular wetland is noisy with a higher inventory value (e.g., $^{137}$Cs profile of S-LO-I-W4-T2-CW-R2 in Supplementary Fig. 2a). In that case, the impact by erosional processes can be deduced with higher certainty. The higher observed inventory value could result from the movement of enriched material via erosion/lateral flow to the center of the wetland, increasing the number of $^{137}$Cs."

Line 416: in the last century?

**Authors' Response:**

We specified the dating technique and its importance in estimating recent OC sequestration rates in the revised manuscript (**Lines 828-829**).

**Original:** "Radiometric dating presents a valuable tool for estimating the OC sequestration potential of wetlands."

**Revised:** "Radiometric dating using $^{137}$Cs and $^{210}$Pb presents a valuable tool for estimating the recent OC sequestration potential of wetlands."

Table 1: Why mention only the 1963 peak if your article is a technical note? Why not present the other peaks (sources of 137Cs) that could be found in other regions of the world? The 1963 peak, which is generally not dated to 1963 in the southern hemisphere? Why not include 241Am, which you may have in your gamma measurements and which may be useful in age modeling? These concepts need to be explained somewhere in your manuscript.

**Authors' Response:**

We recognize the potential for the use of $^{241}$Am as an alternative to $^{137}$Cs; however, $^{241}$Am has similar challenges as $^{210}$Pb measured using gamma spectrometry: $^{241}$Am is detected in the low-energy range of the spectrum where detection is less accurate; i.e., it is more difficult to distinguish the energy peak from the background noise in the spectrum. We are not confident that $^{241}$Am would provide

any greater ability to distinguish 1963 peaks in the sediment profile. The problem we are observing is wide peaks or multiple peaks spanning multiple sample increments, and this problem will exist with either $^{241}$Am or $^{137}$Cs.

Figures 2; 3: Mass or mass depth?

**Authors' Response:**

It should be mass depth. We replaced "mass" with "mass depth" on the x-axis of relevant plots in Figures 2 and 3 in the first submission and 3 and 4 in the revised manuscript.

Table 3: Most of the disadvantages of the 210Pb listed in this table are related to the alpha spectrometry method, but the 210Pb can also be measured by gamma spectrometry, which is easier to use.

**Authors' Response:**

As noted above, we measured $^{210}$Pb by both alpha and gamma spectroscopy.

Since we analyze samples for $^{137}$Cs using gamma spectroscopy, at the same time we prepare samples for $^{210}$Pb using gamma spectroscopy. This requires additional preparation and staging time, and can require more lengthy analysis time, so it is not as simple as getting the two measures at the same time with no extra cost or time.

We find the uncertainty in our $^{210}$Pb results derived from gamma spectroscopy can be unacceptably high, so we often rerun samples for $^{210}$Pb using alpha spectroscopy. Preparation of samples for alpha spectrometry is trickier, but we have the luxury of having the capacity to use both alpha and gamma spectroscopy. If we had to choose between gamma and alpha for $^{210}$Pb, we would choose alpha.

We added the following bullet points in Table 3 to clarify.

**Revised:**

Under advantages of $^{210}$Pb$_{ex}$

- "Less sample preparation time for gamma analysis compared to alpha.

- Gamma analysis is non-destructive, so samples can be re-analyzed for other analyses compared to alpha.

- Can run multiple samples at a time on a single detector in alpha method.

Under disadvantages of $^{210}$Pb$_{ex}$

- Uncertainty of $^{210}$Pb$_{ex}$ results derived from gamma analysis can be higher than alpha."

**Response to comments of reviewer #2:**

This manuscript compares two radiometric techniques as complementary methods for estimating carbon sequestration in wetland soils. The methods for dating using cesium-137 and lead-210 radioisotopes were well explained and the results on the radioisotope profiles were also presented and interpreted well. Given the increasing interest in applying radiometry to the soil carbon sequestration, I think this paper can contribute to disseminating well-documented radiometric methods among the researchers in the various biogeoscience fields. However, I had some concerns about the focus of the current version and a couple of critical methodological details. To be qualified as a technical note, the authors may need to strengthen the "technical" part of the manuscript, and above all technical recommendations readers would expect for this article type. I would also thank the authors if clarify core selection criteria and the limitation of LOI as a measure of soil organic carbon as detailed below.

**Authors' Response:**

Thank you for this feedback. We clarified this in the revised manuscript by elaborating the selection criteria, incorporating technical recommendations at the end of discussion, and providing limitation of LOI as a measure of soil organic carbon (OC). The points mentioned in this are also addressed in a few of our responses to the reviewer's comments below.

<Major comments>

The objective of the technical notes paper

The stated objective ("This research paper compares the use of 137Cs- and 210Pb to estimate recent OC sequestration rates in intact (i.e., not directly 85 impaired by human activities) freshwater mineral soil wetlands located on agricultural landscapes". "This study helps reduce uncertainty in studies that rely on 137Cs or 210Pb radioisotope dating") seems more relevant for regular research articles. Given the article type (technical notes), providing more concrete goals would help readers recognize the main contribution of this paper (e.g., providing specific recommendations for selecting better methods or procedures depending on wetland types). In this context, the last sentence of the abstract, along with some concluding part of the Discussion section (lines 400-413), needs also to be more articulated in providing technical suggestions based on the compared results. In terms of providing technical suggestions, the authors can be more straightforward in specifying which methods are more relevant in which types of samples.

**Authors' Response:**

Thank you for this feedback. We added specific objectives in the Introduction and technical suggestions in the Discussion to ease readability.

**Revised:**

**Lines 155-161:** "The main objective of this research paper is to explore the use of $^{137}Cs$ and $^{210}Pb$ to estimate OC sequestration rates in undisturbed (i.e., not directly impaired by human activities) in freshwater wetlands located on agricultural landscapes. Here, we aim to: (1) categorize $^{137}Cs$ or $^{210}Pb$ profiles into high- and low-quality via a decision framework, (2) apply the decision framework to estimate OC sequestration rates, (3) use 1963 and 1954 time-markers to compare the $^{137}Cs$ and $^{210}Pb$ based OC sequestration rates to

get a better understanding of the sedimentation history, and (4) select the best approach for $^{137}$Cs and $^{210}$Pb to estimate the OC sequestration rates with highest precision."

**Lines 805-825:** "Based on the results of this study, we recommend (1) use high-quality $^{137}$Cs and $^{210}$Pb profiles to estimate OC sequestration rates, (2) interpret $^{137}$Cs profiles from agricultural landscapes carefully from the perspective of redistribution of sediments, (3)use both $^{137}$Cs and $^{210}$Pb to compare and validate estimates if logistic approves. However, in case where one had to choose between $^{137}$Cs and $^{210}$Pb we recommend (1) For $^{137}$Cs: use 1963 time-markers to estimate OC sequestration rates (compared to 1954) since it is found to be most comparable with $^{210}$Pb dating techniques (CFCS model), (2) For $^{210}$Pb (CFCS model): OC sequestration rates from present to 1963 can be estimated with highest precision since we corroborated the estimates with $^{137}$Cs, however, we can not comment on the precision of $^{210}$Pb based OC sequestration rate estimation before 1963 based on the scope of this study."

Criteria for selecting suitable cores

Different criteria were used for selecting suitable profiles of Cs and Pb (Line 121-124). First, more detailed descriptions would help readers understand the rationales of the employed criteria. Second, as the triplicate samples were analyzed for each site, please clarify if the selection criteria were applied to each single profile or an averaged result from three replicates per site. Third, it would provide more quantitative evaluation of the selection if it were clearly stated which proportion of the three profiles (e.g., at least two or all three) fulfill selection requirements.

**Authors' Response:**

Thank you for this feedback. We mentioned in the manuscript that we have first selected the suitable sediment cores (complete and datable) out of the total 90 (30 wetlands × 3 replicates = 90) sediment cores collected from the field. The suitability (complete and datable does not mean high-quality) was assessed by zero activity before the onset and the peak of $^{137}$Cs activity for $^{137}$Cs and was assessed by determining the exponential decline in $^{210}$Pb activity with depth until background levels are reached for $^{210}$Pb. As indicated in the manuscript, we applied the screening for 79 $^{137}$Cs and 47 $^{210}$Pb sediment cores (we did not average). Details on how many replicates were suitably dated and used for statistical analysis can be found in Fig 2 and supplementary figures 2 to 13 of the submitted manuscript.

We clarified this further in the revised manuscript (**Lines 267-271**):

**Original:** "Selecting suitable cores: Of the 90 sediment cores, 79 were suitable (complete and datable) for $^{137}$Cs dating and 47 were suitable for $^{210}$Pb dating. Suitability for $^{137}$Cs profiles for dating was assessed by zero activity before the onset and the peak of $^{137}$Cs activity. The suitability of $^{210}$Pb profiles for dating was assessed by determining the exponential decline in $^{210}$Pb activity with depth until background levels are reached."

**Revised:** "Selecting suitable cores: Of the 90 sediment cores (30 wetlands x 3 replicates = 90), 79 were suitable (complete and datable) for $^{137}$Cs dating and 47 were suitable for $^{210}$Pb dating. Only some replicates from the same wetland were suitable for interpretation or further screening. Suitability for $^{137}$Cs profiles for dating was assessed by zero activity before the onset and the peak of $^{137}$Cs activity. The suitability of $^{210}$Pb profiles for dating was assessed by determining the exponential decline in $^{210}$Pb activity with depth until background levels are reached."

And here as well **(Lines 373-374)**:

**Original:** "Statistical analyses were conducted using sediment cores where both $^{137}$Cs and $^{210}$Pb-based OC sequestration rates were available."

**Revised:** "Statistical analyses were conducted using sediment cores where both $^{137}$Cs and $^{210}$Pb-based OC sequestration rates were available (number of sediment cores (n) = 44)."

The term organic C

The authors used organic C to refer to the organic matter content measured by LOI ("OC content was calculated from OC concentration in % measured by loss-on-ignition method"). LOI is a measure of organic matter, but cannot represent organic carbon quantitatively. The limitation of LOI as a measure of organic matter is well known, such as the ignition of non-organic particles at high temperatures). In my opinion, the results of LOI measurements should be reported as LOI (%). Equating LOI with OC would be unacceptable for many soil scientists. I would recommend measuring and reporting OC. If this is not possible, LOI (%) or OM (%) should be used with a prior definition.

**Authors' Response:**

Thank you for this feedback. We added the suggested definitions and limitations in the revised manuscript (**Lines 364-371**):

**Original:** "OC stocks for the 1954 and 1963 time-markers were calculated by multiplying the OC content per unit mass of soil (g; OC content was calculated from OC concentration in % measured by loss-on-ignition method, Kolthoff and Sandell, 1952) by the mass of sediment for each section interval and specific depth interval per unit area (g cm$^{-2}$) down the profile to the respective time-marker."

**Revised:** "OC stocks for the 1954 and 1963 time-markers were calculated by multiplying the OC content per unit mass of soil (g). Here, OC content was calculated from OC concentration (%) measured by loss-on-ignition (LOI) method (Kolthoff and Sandell, 1952) by the mass of sediment for each section interval and specific depth interval per unit area (g cm$^{-2}$) down the profile to the respective time-marker. OC (%) was calculated by multiplying organic matter (%) by LOI with 0.58 assuming 58% of the organic matter is carbon. Despite the wide applicability, simplicity in measurement techniques, and cost-effectiveness, the LOI approach is associated with some limitations such as ignition of non-organic particles at high temperatures or use of conventional conversion factor (Pribyl, 2010; Hoogsteen et al., 2015) which can result in over-estimation of OC content."

**Citations:**

Pribyl, D. W.: A critical review of the conventional SOC to SOM conversion factor, Geoderma, 156(3-4), 75-83, doi:10.1016/j.geoderma.2010.02.003, 2010.

Hoogsteen, M. J., Lantinga, E. A., Bakker, E. J., Groot, J. C., and Tittonell, P. A.: Estimating soil organic carbon through loss on ignition: effects of ignition conditions and structural water loss, European Journal of soil science, 66(2), 320-328, doi:10.1111/ejss.12224, 2015.

<Minor comments>

Title: if C accumulation occurs mainly in the wetland soils, the title should end like "freshwater wetland soils".

**Authors' Response:**

Title was revised as follows.

**Original:** "Technical Note: Comparison of radiometric techniques for estimating recent organic carbon sequestration rates in freshwater mineral soil wetlands"

**Revised**: "Technical Note: Comparison of radiometric techniques for estimating recent organic carbon sequestration rates in temperate inland wetland soils"

Line (L) 23: Without citing any relevant papers, this sentence assumes that wetlands function as C sinks, although wetlands can also function as C sources under degrading conditions or when CH4 emissions are taken into consideration. Please refine the text based these considerations and provide at least a few relevant papers.

**Authors' Response:**

Sentence was refined, and citation added in the revised manuscript (**Lines 24-27**):

**Original:** "Moreover, these wetlands have the potential to sequester organic carbon (OC), making them candidates to be natural climate solutions by offsetting carbon emissions."

**Revised:** "Moreover, these wetlands have the potential to sequester organic carbon (OC) (Bridgham et al., 2006; Nahlik and Fennessey, 2016; Bansal et al., 2023). Accounting for the balance between the sequestration and emission of carbon can help establish wetlands as important candidates for natural climate solutions by offsetting carbon emissions (Hamback et al., 2023)."

**Citations:**

Bansal, S., Creed, I. F., Tangen, B. A., Bridgham, S. D., Desai, A. R., Krauss, K. W., Neubauer, S. C., Noe, G. B., Rosenberry, D. O., Trettin, C., Wickland, K. P., Allen, S. T., Arias-Ortiz, A., Armitage, A. R., Baldocchi, D., Banerjee, K., Bastviken, D., Berg, P., Bogard, M., Chow, A. T., Conner, W. H., Craft, C., Creamer, C., DelSontro, T., Duberstein, J. A., Eagle, M., Fennessy, M. S., Finkelstein, S. A., Göckede, M., Grunwald, S., Halabisky, M., Herbert, E., Jahangir, M. M. R., Johnson, O. F., Jones, M. C., Kelleway, J. J., Knox, S., Kroeger, K. D., Kuehn, K. A., Lobb, D., Loder A. L., Ma, S., Maher, D. T., McNicol, G., Meier, J., Middleton, B. A., Mills, C., Mistry, P., Mitra, A., Mobiian, C., Nahlik, A. M., Newman, S., O'Connell, J. L., Oikawa, P., Post van der Burg, M., Schutte, C. A., Song, C., Stagg, C. L., Turner, J., Vargas, R., Waldrop, M. P., Wallin, M. B., Wang, Z. A., Ward, E. J., Willard, D. A., Yarwood, S., and Zhu X.: Practical guide to measuring wetland carbon pools and fluxes, Wetlands, 43(8), 105, doi:10.1007/s13157-023-01722-2, 2023.

Bridgham, S. D., Megonigal, J. P., Keller, J. K., Bliss, N. B., and Trettin, C.: The carbon balance of North American wetlands, Wetlands, 26(4), 889-916, doi:10.1672/0277-5212(2006)26[889:TCBONA]2.0.CO;2, 2006.

Hambäck, P. A., Dawson, L., Geranmayeh, P., Jarsjö, J., Kačergytė, I., Peacock, M., Collentine, D., Destouni, G., Futter, M., Hugelius, G., Hedman, S., Jonsson, S., Klatt, B. K., Lindström, A., Nilsson, J. E., Pärt, T., Schneider, L. D., Strand, J. A., Urrutia-Cordero, P., Åhlén, D., Åhlén., I., and Blicharska, M.: Tradeoffs and synergies in wetland multifunctionality: A scaling issue. Science of the Total Environment, 862, 160746, doi:10.1016/j.scitotenv.2022.160746, 2023.

Nahlik, A. M. and Fennessy, M. S.: Carbon storage in US wetlands. Nature Communications, 7(1), 1-9, doi:10.1038/ncomms13835, 2016.

L 56 "210Pb is a naturally occurring radionuclide of 238U": Do you mean "…radionuclide deriving from 238U?"

**Authors' Response:**

Yes. Sentence was refined for readability (**Lines 98-99**).

**Original:** "Unlike $^{137}$Cs, $^{210}$Pb is a naturally occurring radionuclide of $^{238}$U and deposits atmospherically from the decay of 226Ra (Walling and He, 1999)."

**Revised:** "Unlike $^{137}$Cs, $^{210}$Pb is a naturally occurring radionuclide derived from $^{238}$U and deposits atmospherically from the decay of $^{226}$Ra (Walling and He, 1999)."

L 66: It would be more reader-friendly if you explain "unsupported" and "supported" by adding a few more words.

**Authors' Response:**

Following text was added to explain "unsupported" and "supported" (**Lines 107-112**):

**Original:** "Gamma and alpha spectrometry of $^{210}$Pb provides the total $^{210}$Pb activity, which incorporates unsupported (or excess) $^{210}$Pb ($^{210}$Pb$_{ex}$) and supported $^{210}$Pb."

**Revised:** "Gamma and alpha spectrometry of $^{210}$Pb provides the total $^{210}$Pb activity, which incorporates unsupported (or excess) $^{210}$Pb ($^{210}$Pb$_{ex}$) and supported $^{210}$Pb. Supported $^{210}$Pb is derived from the natural decay of radium-226 ($^{226}$Ra) present in the sediment while unsupported $^{210}$Pb comes from the decay of atmospheric radon-222 ($^{222}$Rn), which deposits $^{210}$Pb onto the sediment surface from the air. Unsupported $^{210}$Pb activity decreases over time due to radioactive decay, unlike supported $^{210}$Pb (Appleby and Oldfieldz, 1983)."

**Citation:**

Appleby, P. G., and Oldfieldz, F.: The assessment of 210 Pb data from sites with varying sediment accumulation rates, Hydrobiologia, 103, 29-35, doi:10.1007/BF00028424, 1983.

L 76: Please remove "according to" and instead put the references in parentheses.

**Authors' Response:**

"According to" was removed and references put in parentheses.

L 87: Parentheses here appear unnecessary.

**Authors' Response:**

Parentheses were removed.

L 108-108 "the high-purity germanium detectors, Broad Energy Germanium detectors (BE6530) and high-resolution Small Anode Germanium well detectors (GSW275L) (Mirion Technologies, Inc., Atlanta, GA, USA)": It is not clear whether BE6530 and GSW275L are high-purity and high-resolution detectors, respectively.

**Authors' Response:**

Sentence was refined to clarify both BE6530 and GSW275L are high-purity detectors (**Lines: 225-228**):

**Original:** "The gamma analysis was conducted using the high-purity germanium detectors, Broad Energy Germanium detectors (BE6530) and high-resolution Small Anode Germanium well detectors (GSW275L) (Mirion Technologies, Inc., Atlanta, GA, USA). The alpha analysis was conducted using ORTEC® alpha spectrometer (AMETEK® Advanced Measurement Technology, TN, USA)"

**Revised:** "The gamma analysis was conducted using the high-purity germanium detectors; e.g., Broad Energy Germanium detectors (BE6530) and Small Anode Germanium well detectors (GSW275L) (Mirion Technologies, Inc., Atlanta, GA, USA). The alpha analysis was conducted using ORTEC® alpha spectrometer (AMETEK® Advanced Measurement Technology, TN, USA)"

L 112: This would be a good place to describe any QA/QC measures that were employed to guarantee the analytical accuracy.

**Authors' Response:**

We added the following text in the revised manuscript (**Lines 231-235**):

"Measurement accuracy of gamma detectors is ensured by assessing the counting errors with reference materials within same geometry as the sample (e.g., petri dish). Detection error was < 10% with a counting time of up to 24 h. Furthermore, Landscape Dynamics Laboratory undergoes regular Proficiency Testing through the International Atomic Reference Material Agency (IARMA) and previously through the International Atomic Energy Agency (IAEA) to ensure acceptable accuracy and precision of analytical results using gamma spectroscopy."

L 232-238: Don't these exceptions demand a refinement of the criteria? For instance, additional criteria can be prescribed considering these exceptions. It would be too arbitrary if we have to accept the failed profiles based on subjective visual inspections.

**Authors' Response:**

These exceptions were reflected in the currently set criteria, where the first step is to look at the shape of the $^{137}$Cs peak: "clear peaks with 2 or 3 points on both sides of the peak". In the absence of clear peaks, the cumulative $^{137}$Cs inventory value should be checked. We wanted to elaborate on this criterion (and how we followed it) by providing explanation for two cores where, despite a cumulative $^{137}$Cs inventory value < 500 Bq m$^{-2}$ we considered them as high-quality because the 1963 peak was good. For one sediment core, we did take a subjective decision by choosing it to be high-quality rather than low-quality. We acknowledge that a classification of the three categories instead of two would have been better in the case where we can classify the cores into low, medium- or high-quality. However, with limited number of sediment cores and to promote ease of applying the decision framework, we restricted ourselves to two categories. Based on the response to the other comments below, we also provided examples of which profile we are referring to throughout the revised manuscript so that the reader can refer to the specific $^{137}$Cs profiles in the main manuscript or supplementary figures. An example of the revised text is provided below (**Lines 445-447**):

**Original:** "Two $^{137}$Cs profiles were considered high-quality despite a cumulative $^{137}$Cs inventory value < 500 Bq m$^{-2}$ because the 1963 peak was clear, distinct, and elongated with two-to-three points on both sides of the peak."

**Revised:** "Two $^{137}$Cs profiles were considered high-quality despite a cumulative $^{137}$Cs inventory value < 500 Bq m-2 because the 1963 peak was clear, distinct, and elongated with two-to-three points on both sides of the peak (e.g., $^{137}$Cs profile of M-OA-I-W4-T2-CW-R2 in Supplementary Fig. 7b)."

L 306: Please include a phrase mentioning OC data like "combined with OC measurements" following "both radioisotopes dating".

**Authors' Response:**

We did not include the phrase "combined with OC measurements" since we did not compare OC measurements or stocks or contents in this study. Here, we focused on comparing the $^{137}$Cs and $^{210}$Pb based OC sequestration rates where calculations include OC stock measurements. Details on how we calculated the OC sequestration rates/OC stock can be found in the Methods section.

L 256-261: Please combine these sentences into a single combed paragraph. The following 2-3 sentences throughout the Results section also seem untidy, requiring some editorial refinement.

**Authors' Response:**

Line 256 – 258 were deleted in the revised manuscript based on a reviewer #1 comment. The paragraph now starts with "For each of the ……." as shown below. We revised the text as suggested (**Lines 488-489**).

**Original:** "The comparability of $^{137}$Cs vs. $^{210}$Pb derived OC sequestration rates was investigated through both visual inspection of the Q-Q plots and the Cramer-von Mises test which assigned significance of the distance of the points from the 1:1 line assessed with p-value and the AIC. For each of the four datasets (D1-D4), the points on the Q-Q plot were distributed in a straight line, showing a linear relationship between the two estimates being compared (R2 > 0.95, p-value < 0.001) (Fig. 4)."

**Revised:** "For each of the four datasets (D1-D4), the points on the Q-Q plot were distributed in a straight line, showing a linear relationship between the two estimates being compared (R2 > 0.95, p-value < 0.001) (Fig. 5)."

L 383-390: Given the article type (technical note) and the goal of suggesting good practices, it would be helpful if you provide some recommendations about the highly disturbed cases.

**Authors' Response:**

Thanks for the feedback. Throughout the manuscript we tried to provide recommendations regarding highly disturbed cases. For e.g., we mentioned that if the $^{137}$Cs profiles are noisy with a higher inventory value, then the impact by erosional processes can be deduced with higher certainty because the higher observed inventory value could be a result of movement of enriched material to the center of the wetland, therefore increasing the quantity of $^{137}$Cs from the value that would be expected if no new enriched material was introduced via erosion/lateral flow. Therefore, these $^{137}$Cs profiles should not be discarded due to its noise. In the case where there are two peaks seen with higher inventory, one can reposition the $^{137}$Cs peak to account for enrichment as we did for few cores. We provided examples of which $^{137}$Cs or $^{210}$Pb profile we are referring to throughout the revised manuscript so that the reader can refer to the figures/profiles in the main manuscript or supplement to visualize the disturbed cores/profiles. An example of how we will do revisions is provided below (**Lines 726-729**):

**Original:** "If the $^{137}$Cs activity of most of the sediment cores from an individual wetland are noisy with a higher inventory value, then the impact by erosional processes can be deduced with higher certainty because the higher observed inventory value could be a result of movement of enriched material to the center of the wetland, therefore increasing the quantity of $^{137}$Cs from the value that would be expected if no new enriched material was introduced via erosion/lateral flow."

**Revised:** "Suppose the 137Cs activity of most of the sediment cores from a particular wetland is noisy with a higher inventory value (e.g., 137Cs profile of S-LO-I-W4-T2-CW-R2 in Supplementary Fig. 2a). In that case, the impact by erosional processes can be deduced with higher certainty. The higher observed inventory value could result from the movement of enriched material via erosion/lateral flow to the center of the wetland, increasing the number of 137Cs."

L 394: Please cite the relevant figure.

**Authors' Response:**

Relevant figure was cited in the revised manuscript.

L 395-398: This assessment is not fully in line with the results shown in Table 2 (particularly the four mean values of C sequestration rates) and the main conclusion on the compatibility of both methods. Don't you need to mention at least the results (Table 2) here?

**Authors' Response:**

We think the results shown in Table 2 align with the main conclusion of this study, i.e., $^{137}$Cs and $^{210}$Pb based OC sequestration rates since 1963 using high-quality profiles are reasonably comparable or similar (for e.g., compared to using 1954 time-marker). We referred to Table 2 results in the revised text (**Lines 781-789**):

**Original:** "Conversely, the $^{137}$Cs-$^{210}$Pb Q-Q plot of the 1954 OC sequestration rates showed more deviation from the 1:1 line; $^{137}$Cs-based OC sequestration rates were more dispersed and were higher than the $^{210}$Pb-based OC sequestration rates. Providing better

sequestration rate estimates has consequences for estimating OC stocks with an improved degree of accuracy, which may provide policymakers with better tools to make informed carbon management decisions supported with data."

**Revised:** "Conversely, the [137]Cs-[210]Pb Q-Q plot of the 1954 OC sequestration rates showed more deviation from the 1:1 line; [137]Cs-based OC sequestration rates were more dispersed and were higher than the [210]Pb-based OC sequestration rates (Fig 5a and 5b). The mean OC sequestration rates in Table 2 further verify the comparability of OC sequestration rates using 1963 time-marker (mean [137]Cs OC sequestration rate is 0.63 Mg ha$^{-1}$ yr$^{-1}$ while mean [210]Pb OC sequestration rate is 0.68 Mg ha$^{-1}$ yr$^{-1}$) and the dispersion using 1954 time-marker (mean [137]Cs OC sequestration rate is 1.02 Mg ha$^{-1}$ yr$^{-1}$ while mean [210]Pb OC sequestration rate is 0.67 Mg ha$^{-1}$ yr$^{-1}$). Providing better sequestration rate estimates has consequences for estimating OC stocks with an improved degree of accuracy, which may provide policymakers with better tools to make informed carbon management decisions supported with data."

**Important note:** We will update the wetland sediment core id in the revised manuscript and supplement to maintain consistency in future publication. Please refer to the table below for the previous codes and the revised codes.

| Previously used wetland sediment core ID in the submitted technical note | Revised wetland sediment core ID in the revised manuscript |
|---|---|
| ABB1T1W1 | A-BE-R4-W4-T1-CW-R1 |
| ABB1T2W1 | A-BE-R4-W4-T2-CW-R2 |
| ABB1T3W1 | A-BE-R4-W4-T3-CW-R3 |
| AMB1T1W1 | A-WE-R23-W24-T1-CW-R1 |
| AMB1T2W1 | A-WE-R23-W24-T2-CW-R2 |
| AMB1T3W1 | A-WE-R23-W24-T3-CW-R3 |
| BER1T1W1 | A-CA-R7-W8-T1-CW-R1 |
| BER1T2W1 | A-CA-R7-W8-T2-CW-R2 |
| BER1T3W1 | A-CA-R7-W8-T3-CW-R3 |
| BOW1T1W1 | A-CA-R4-W5-T1-CW-R1 |
| BOW1T2W1 | A-CA-R4-W5-T2-CW-R2 |
| BOW1T3W1 | A-CA-R4-W5-T3-CW-R3 |
| BOW2T1W1 | A-CA-R11-W14-T1-CW-R1 |
| BOW2T2W1 | A-CA-R11-W14-T2-CW-R2 |
| BOW2T3W1 | A-CA-R11-W14-T3-CW-R3 |
| BOW3T1W1 | A-CA-R11-W15-T1-CW-R1 |
| BOW3T2W1 | A-CA-R11-W15-T2-CW-R2 |
| BOW3T3W1 | A-CA-R11-W15-T3-CW-R3 |
| BUS1T1W1 | A-CA-R7-W9-T1-CW-R1 |
| BUS1T2W1 | A-CA-R7-W9-T2-CW-R2 |
| BUS1T3W1 | A-CA-R7-W9-T3-CW-R3 |
| CHR1T1W1 | A-BE-R10-W13-T1-CW-R1 |
| CHR1T2W1 | A-BE-R10-W13-T2-CW-R2 |
| CHR1T3W1 | A-BE-R10-W13-T3-CW-R3 |
| CUR1T1W1 | A-CA-D-W1-T1-CW-R1 |
| CUR1T2W1 | A-CA-D-W1-T2-CW-R2 |
| CUR1T3W1 | A-CA-D-W1-T3-CW-R3 |
| CUR2T1W1 | A-CA-D-W2-T1-CW-R1 |
| CUR2T2W1 | A-CA-D-W2-T2-CW-R2 |
| CUR2T3W1 | A-CA-D-W2-T3-CW-R3 |
| CUR3T1W1 | A-CA-D-W3-T1-CW-R1 |
| CUR3T2W1 | A-CA-D-W3-T2-CW-R2 |
| CUR3T3W1 | A-CA-D-W3-T3-CW-R3 |
| FER1T1W1 | A-LA-R14-W16-T1-CW-R1 |
| FER1T2W1 | A-LA-R14-W16-T2-CW-R2 |
| FER1T3W1 | A-LA-R14-W16-T3-CW-R3 |
| FER2T1W1 | A-LA-R14-W17-T1-CW-R1 |
| FER2T2W1 | A-LA-R14-W17-T2-CW-R2 |
| FER2T3W1 | A-LA-R14-W17-T3-CW-R3 |
| FOR1T1W1 | A-WE-R3-W2-T1-CW-R1 |
| FOR1T2W1 | A-WE-R3-W2-T2-CW-R2 |
| FOR1T3W1 | A-WE-R3-W2-T3-CW-R3 |
| FOR2T1W1 | A-WE-R2-W1-T1-CW-R1 |
| FOR2T2W1 | A-WE-R2-W1-T2-CW-R2 |
| FOR2T3W1 | A-WE-R2-W1-T3-CW-R3 |
| HEN1T1W1 | A-BE-R9-W11-T1-CW-R1 |
| HEN1T2W1 | A-BE-R9-W11-T2-CW-R2 |
| HEN1T3W1 | A-BE-R9-W11-T3-CW-R3 |
| INT1T1W1 | A-WE-I-W1-T1-CW-R1 |
| INT1T2W1 | A-WE-I-W1-T2-CW-R2 |
| INT1T3W1 | A-WE-I-W1-T3-CW-R3 |
| INT2T1W1 | A-WE-I-W2-T1-CW-R1 |

| Previously used wetland sediment core ID in the submitted technical note | Revised wetland sediment core ID in the revised manuscript |
| --- | --- |
| INT2T2W1 | A-WE-I-W2-T2-CW-R2 |
| INT2T3W1 | A-WE-I-W2-T3-CW-R3 |
| INT3T1W1 | A-CA-I-W3-T1-CW-R1 |
| INT3T2W1 | A-CA-I-W3-T2-CW-R2 |
| INT3T3W1 | A-CA-I-W3-T3-CW-R3 |
| KEM1T1W1 | A-RE-R16-W19-T1-CW-R1 |
| KEM1T2W1 | A-RE-R16-W19-T2-CW-R2 |
| KEM1T3W1 | A-RE-R16-W19-T3-CW-R3 |
| KEM2T1W1 | A-RE-R16-W20-T1-CW-R1 |
| KEM2T2W1 | A-RE-R16-W20-T2-CW-R2 |
| KEM2T3W1 | A-RE-R16-W20-T3-CW-R3 |
| LAB1T1W1 | A-WE-R6-W7-T1-CW-R1 |
| LAB1T2W1 | A-WE-R6-W7-T2-CW-R2 |
| LAB1T3W1 | A-WE-R6-W7-T3-CW-R3 |
| MAR1T1W1 | A-CA-R22-W22-T1-CW-R1 |
| MAR1T2W1 | A-CA-R22-W22-T2-CW-R2 |
| MAR1T3W1 | A-CA-R22-W22-T3-CW-R3 |
| MCN1T1W1 | A-CA-R14-W18-T1-CW-R1 |
| MCN1T2W1 | A-CA-R14-W18-T2-CW-R2 |
| MCN1T3W1 | A-CA-R14-W18-T3-CW-R3 |
| MIT1T1W1 | A-CA-R22-W23-T1-CW-R1 |
| MIT1T2W1 | A-CA-R22-W23-T2-CW-R2 |
| MIT1T3W1 | A-CA-R22-W23-T3-CW-R3 |
| NAS1T1W1 | A-BE-R8-W10-T1-CW-R1 |
| NAS1T2W1 | A-BE-R8-W10-T2-CW-R2 |
| NAS1T3W1 | A-BE-R8-W10-T3-CW-R3 |
| OZM1T1W1 | A-CA-R5-W6-T1-CW-R1 |
| OZM1T2W1 | A-CA-R5-W6-T2-CW-R2 |
| OZM1T3W1 | A-CA-R5-W6-T3-CW-R3 |
| RAU1T1W1 | A-BE-R21-W21-T1-CW-R1 |
| RAU1T2W1 | A-BE-R21-W21-T2-CW-R2 |
| RAU1T3W1 | A-BE-R21-W21-T3-CW-R3 |
| REU1T1W1 | A-BE-R9-W12-T1-CW-R1 |
| REU1T2W1 | A-BE-R9-W12-T2-CW-R2 |
| REU1T3W1 | A-BE-R9-W12-T3-CW-R3 |
| ROP1T1W1 | A-CA-R3-W3-T1-CW-R1 |
| ROP1T2W1 | A-CA-R3-W3-T2-CW-R2 |
| ROP1T3W1 | A-CA-R3-W3-T3-CW-R3 |
| WETAT1W1 | M-OA-I-W1-T1-CW-R1 |
| WETAT2W1 | M-OA-I-W1-T2-CW-R2 |
| WETAT3W1 | M-OA-I-W1-T3-CW-R3 |
| WETET1W1 | M-OA-I-W2-T1-CW-R1 |
| WETET2W1 | M-OA-I-W2-T2-CW-R2 |
| WETET3W1 | M-OA-I-W2-T3-CW-R3 |
| WETJT1W1 | M-OA-I-W3-T1-CW-R1 |
| WETJT2W1 | M-OA-I-W3-T2-CW-R2 |
| WETJT3W1 | M-OA-I-W3-T3-CW-R3 |
| WETKT1W1 | M-OA-I-W4-T1-CW-R1 |
| WETKT2W1 | M-OA-I-W4-T2-CW-R2 |
| WETKT3W1 | M-OA-I-W4-T3-CW-R3 |
| WETXT1W1 | M-OA-I-W5-T1-CW-R1 |
| WETXT2W1 | M-OA-I-W5-T2-CW-R2 |
| WETXT3W1 | M-OA-I-W5-T3-CW-R3 |
| ADB1T1W1 | O-AL-D-W4-T1-CW-R1 |
| ADB1T2W1 | O-AL-D-W4-T2-CW-R2 |
| ADB1T3W1 | O-AL-D-W4-T3-CW-R3 |

| Previously used wetland sediment core ID in the submitted technical note | Revised wetland sediment core ID in the revised manuscript |
| --- | --- |
| ADB2T1W1 | O-AL-D-W5-T1-CW-R1 |
| ADB2T2W1 | O-AL-D-W5-T2-CW-R2 |
| ADB2T3W1 | O-AL-D-W5-T3-CW-R3 |
| ADB3T1W1 | O-AL-D-W6-T1-CW-R1 |
| ADB3T2W1 | O-AL-D-W6-T2-CW-R2 |
| ADB3T3W1 | O-AL-D-W6-T3-CW-R3 |
| AI-1T1W1 | O-AL-I-W4-T1-CW-R1 |
| AI-1T2W1 | O-AL-I-W4-T2-CW-R2 |
| AI-1T3W1 | O-AL-I-W4-T3-CW-R3 |
| AI-2T1W1 | O-AL-I-W5-T1-CW-R1 |
| AI-2T2W1 | O-AL-I-W5-T2-CW-R2 |
| AI-2T3W1 | O-AL-I-W5-T3-CW-R3 |
| AI-3T1W1 | O-AL-I-W6-T1-CW-R1 |
| AI-3T2W1 | O-AL-I-W6-T2-CW-R2 |
| AI-3T3W1 | O-AL-I-W6-T3-CW-R3 |
| AR12-1T1W1 | O-AL-R18-W13-T1-CW-R1 |
| AR12-1T2W1 | O-AL-R18-W13-T2-CW-R2 |
| AR12-1T3W1 | O-AL-R18-W13-T3-CW-R3 |
| AR12-2T1W1 | O-AL-R18-W14-T1-CW-R1 |
| AR12-2T2W1 | O-AL-R18-W14-T2-CW-R2 |
| AR12-2T3W1 | O-AL-R18-W14-T3-CW-R3 |
| AR12-3T1W1 | O-AL-R18-W15-T1-CW-R1 |
| AR12-3T2W1 | O-AL-R18-W15-T2-CW-R2 |
| AR12-3T3W1 | O-AL-R18-W15-T3-CW-R3 |
| AR3-1T1W1 | O-AL-R9-W10-T1-CW-R1 |
| AR3-1T2W1 | O-AL-R9-W10-T2-CW-R2 |
| AR3-1T3W1 | O-AL-R9-W10-T3-CW-R3 |
| AR3-2T1W1 | O-AL-R9-W11-T1-CW-R1 |
| AR3-2T2W1 | O-AL-R9-W11-T2-CW-R2 |
| AR3-2T3W1 | O-AL-R9-W11-T3-CW-R3 |
| AR3-3T1W1 | O-AL-R9-W12-T1-CW-R1 |
| AR3-3T2W1 | O-AL-R9-W12-T2-CW-R2 |
| AR3-3T3W1 | O-AL-R9-W12-T3-CW-R3 |
| OD-1T1W1 | O-BR-D-W1-T1-CW-R1 |
| OD-1T2W1 | O-BR-D-W1-T2-CW-R2 |
| OD-1T3W1 | O-BR-D-W1-T3-CW-R3 |
| OD-2T1W1 | O-BR-D-W2-T1-CW-R1 |
| OD-2T2W1 | O-BR-D-W2-T2-CW-R2 |
| OD-2T3W1 | O-BR-D-W2-T3-CW-R3 |
| OD-2T4W1 | O-BR-D-W2-T4-CW-R4 |
| OD-3T1W1 | O-BR-D-W3-T1-CW-R1 |
| OD-3T2W1 | O-BR-D-W3-T2-CW-R2 |
| OD-3T3W1 | O-BR-D-W3-T3-CW-R3 |
| OD-3T4W1 | O-BR-D-W3-T4-CW-R4 |
| OI-1T1W1 | O-BR-I-W1-T1-CW-R1 |
| OI-1T2W1 | O-BR-I-W1-T2-CW-R2 |
| OI-1T3W1 | O-BR-I-W1-T3-CW-R3 |
| OI-4T1W1 | O-BR-I-W2-T1-CW-R1 |
| OI-4T2W1 | O-BR-I-W2-T2-CW-R2 |
| OI-4T3W1 | O-BR-I-W2-T3-CW-R3 |
| OI-4T4W1 | O-BR-I-W2-T4-CW-R4 |
| OI-6T1W1 | O-BR-I-W3-T1-CW-R1 |
| OI-6T2W1 | O-BR-I-W3-T2-CW-R2 |
| OI-6T3W1 | O-BR-I-W3-T3-CW-R3 |
| OR10-1T1W1 | O-BR-R15-W1-T1-CW-R1 |
| OR10-1T2W1 | O-BR-R15-W1-T2-CW-R2 |

| Previously used wetland sediment core ID in the submitted technical note | Revised wetland sediment core ID in the revised manuscript |
| --- | --- |
| OR10-1T3W1 | O-BR-R15-W1-T3-CW-R3 |
| OR10-2T1W1 | O-BR-R15-W2-T1-CW-R1 |
| OR10-2T2W1 | O-BR-R15-W2-T2-CW-R2 |
| OR10-2T3W1 | O-BR-R15-W2-T3-CW-R3 |
| OR10-3T1W1 | O-BR-R15-W3-T1-CW-R1 |
| OR10-3T2W1 | O-BR-R15-W3-T2-CW-R2 |
| OR10-3T3W1 | O-BR-R15-W3-T3-CW-R3 |
| OR20-1T1W1 | O-BR-R25-W4-T1-CW-R1 |
| OR20-1T2W1 | O-BR-R25-W4-T2-CW-R2 |
| OR20-1T3W1 | O-BR-R25-W4-T3-CW-R3 |
| OR20-2T1W1 | O-BR-R25-W5-T1-CW-R1 |
| OR20-2T2W1 | O-BR-R25-W5-T2-CW-R2 |
| OR20-2T3W1 | O-BR-R25-W5-T3-CW-R3 |
| OR20-2T4W1 | O-BR-R25-W5-T4-CW-R4 |
| OR20-3T1W1 | O-BR-R25-W6-T1-CW-R1 |
| OR20-3T2W1 | O-BR-R25-W6-T2-CW-R2 |
| OR20-3T3W1 | O-BR-R25-W6-T3-CW-R3 |
| OR35-1T1W1 | O-BR-R40-W7-T1-CW-R1 |
| OR35-1T2W1 | O-BR-R40-W7-T2-CW-R2 |
| OR35-1T3W1 | O-BR-R40-W7-T3-CW-R3 |
| OR35-2T1W1 | O-BR-R40-W8-T1-CW-R1 |
| OR35-2T2W1 | O-BR-R40-W8-T2-CW-R2 |
| OR35-2T3W1 | O-BR-R40-W8-T3-CW-R3 |
| OR35-3T1W1 | O-BR-R40-W9-T1-CW-R1 |
| OR35-3T2W1 | O-BR-R40-W9-T2-CW-R2 |
| OR35-3T3W1 | O-BR-R40-W9-T3-CW-R3 |
| FAYE1AT1W1 | S-FO-I-W8-T1-CW-R1 |
| FAYE1AT2W1 | S-FO-I-W8-T2-CW-R2 |
| FAYE1AT3W1 | S-FO-I-W8-T3-CW-R3 |
| FAYE1BT1W1 | S-FO-I-W9-T1-CW-R1 |
| FAYE1BT2W1 | S-FO-I-W9-T2-CW-R2 |
| FAYE1BT3W1 | S-FO-I-W9-T3-CW-R3 |
| FAYE2AT1W1 | S-FO-I-W10-T1-CW-R1 |
| FAYE2AT2W1 | S-FO-I-W10-T2-CW-R2 |
| FAYE2AT3W1 | S-FO-I-W10-T3-CW-R3 |
| FAYE2BT1W1 | S-FO-I-W11-T1-CW-R1 |
| FAYE2BT2W1 | S-FO-I-W11-T2-CW-R2 |
| FAYE2BT3W1 | S-FO-I-W11-T3-CW-R3 |
| FAYE3TT1W1 | S-FO-I-W12-T1-CW-R1 |
| FAYE3TT2W1 | S-FO-I-W12-T2-CW-R2 |
| FAYE3TT3W1 | S-FO-I-W12-T3-CW-R3 |
| FAYE4AT1W1 | S-EM-I-W13-T1-CW-R1 |
| FAYE4AT2W1 | S-EM-I-W13-T2-CW-R2 |
| FAYE4AT3W1 | S-EM-I-W13-T3-CW-R3 |
| FAYE4BT1W1 | S-EM-I-W14-T1-CW-R1 |
| FAYE4BT2W1 | S-EM-I-W14-T2-CW-R2 |
| FAYE4BT3W1 | S-EM-I-W14-T3-CW-R3 |
| FAYE5TT1W1 | S-FO-I-W15-T1-CW-R1 |
| FAYE5TT2W1 | S-FO-I-W15-T2-CW-R2 |
| FAYE5TT3W1 | S-FO-I-W15-T3-CW-R3 |
| FAYE6CT1W1 | S-FO-I-W16-T1-CW-R1 |
| FAYE6CT2W1 | S-FO-I-W16-T2-CW-R2 |
| FAYE6CT3W1 | S-FO-I-W16-T3-CW-R3 |
| RIN1TT1W1 | S-RO-I-W1-T1-CW-R1 |
| RIN1TT2W1 | S-RO-I-W1-T2-CW-R2 |
| RIN1TT3W1 | S-RO-I-W1-T3-CW-R3 |

| Previously used wetland sediment core ID in the submitted technical note | Revised wetland sediment core ID in the revised manuscript |
| --- | --- |
| RIN2CT1W1 | S-RO-I-W2-T1-CW-R1 |
| RIN2CT2W1 | S-RO-I-W2-T2-CW-R2 |
| RIN2CT3W1 | S-RO-I-W2-T3-CW-R3 |
| RIN3TT1W1 | S-LO-I-W3-T1-CW-R1 |
| RIN3TT2W1 | S-LO-I-W3-T2-CW-R2 |
| RIN3TT3W1 | S-LO-I-W3-T3-CW-R3 |
| RIN4TT1W1 | S-LO-I-W4-T1-CW-R1 |
| RIN4TT2W1 | S-LO-I-W4-T2-CW-R2 |
| RIN4TT3W1 | S-LO-I-W4-T3-CW-R3 |
| RIN5AT1W1 | S-RO-I-W5-T1-CW-R1 |
| RIN5AT2W1 | S-RO-I-W5-T2-CW-R2 |
| RIN5AT3W1 | S-RO-I-W5-T3-CW-R3 |
| RIN5BT1W1 | S-RO-I-W6-T1-CW-R1 |
| RIN5BT2W1 | S-RO-I-W6-T2-CW-R2 |
| RIN5BT3W1 | S-RO-I-W6-T3-CW-R3 |
| RIN6CT1W1 | S-RU-I-W7-T1-CW-R1 |
| RIN6CT2W1 | S-RU-I-W7-T2-CW-R2 |
| RIN6CT3W1 | S-RU-I-W7-T3-CW-R3 |

---

## Author Response (AR2)

**Response to Reviewer's comments:**

I thank the authors for their important work in revising this article.
Their detailed responses to my questions are convincing, and they have all been answered.

**Authors' Response:**

Thank you for the feedback to improve the quality of the manuscript.

I would suggest that the authors add the data from this article to an open database (e.g. zenodo) where the data will be more easily accessible to the community, rather than in data reports.

**Authors' Response:**

We have uploaded the relevant data in an open database as suggested needed to reproduce the work of this manuscript. We have added the following sentence under "Code and data availability" of the revised manuscript to provide the link for the dataset. The dataset contains a README file along with the dataset containing information on the 44 sediment cores used in this study including, ID, the geographical location, the year of sampling, classification of $^{137}$Cs and $^{210}$Pb profiles, cumulative $^{137}$Cs inventory, and organic carbon sequestration rate since 1954 and 1963 based on $^{137}$Cs and $^{210}$Pb dating techniques.

"The R code for the distance sampling modelling along with the data to run the code is available at https://doi.org/10.5281/zenodo.10951658. The organic carbon (OC) sequestration rates data used to check the comparability of the radioisotope profiles can be found in the Supplement. These sequestration rate data and the geographical locations, years of sampling, and additional information about the sediment cores are available at https://doi.org/10.5281/zenodo.13696300."

Also, it might be interesting to add a geographical term (e.g. North America) in the title to balance the discussion related to the specificity of 137Cs in certain regions of the world.

**Authors' Response:**

Thank you for your suggestion. We think the same technique can be applied anywhere with $^{137}$Cs fallout and opted to stay with the current title. We acknowledge the distribution of $^{137}$Cs is not uniform globally. Therefore, (1) cumulative $^{137}$Cs inventory value to screen/interpret the profiles needs to be validated against the known local reference level, and (2) $^{137}$Cs with additional time-markers, for example in Europe and Japan, need to be adjusted to compare with $^{210}$Pb, but the steps and interpretation outlined in the manuscript can be followed.

As users of fallout radionuclides, we are aware of the regional nature of fallout rates. All users of FRNs are also aware of this. Therefore, we do not think that it is necessary to state "North America" in the title.

Note that we have revised the title instead to "Comparison of radiometric techniques for estimating recent organic carbon sequestration rates in inland wetland soils". That is, we removed the word "temperate" before "inland wetland soils"; the regionality has to do with where the bombs exploded and atmospheric circulation at those locations and times, not the climate.